# EMBO *reports*

# Multiple C2 domains and transmembrane region proteins (MCTPs) tether membranes at plasmodesmata

Marie L Brault[1,¶], Jules D Petit[1,2,¶] ID, Françoise Immel[1] ID, William J Nicolas[1,†] ID, Marie Glavier[1] ID, Lysiane Brocard[3], Amèlia Gaston[1,‡] ID, Mathieu Fouché[1,‡], Timothy J Hawkins[4], Jean-Marc Crowet[2,§] ID, Magali S Grison[1] ID, Véronique Germain[1], Marion Rocher[1], Max Kraner[5], Vikram Alva[6] ID, Stéphane Claverol[7], Andrea Paterlini[8] ID, Ykä Helariutta[8] ID, Magali Deleu[2] ID, Laurence Lins[2] ID, Jens Tilsner[9,10,*] ID & Emmanuelle M Bayer[1,**] ID

## Abstract

In eukaryotes, membrane contact sites (MCS) allow direct communication between organelles. Plants have evolved a unique type of MCS, inside intercellular pores, the plasmodesmata, where endoplasmic reticulum (ER)–plasma membrane (PM) contacts coincide with regulation of cell-to-cell signalling. The molecular mechanism and function of membrane tethering within plasmodesmata remain unknown. Here, we show that the multiple C2 domains and transmembrane region protein (MCTP) family, key regulators of cell-to-cell signalling in plants, act as ER-PM tethers specifically at plasmodesmata. We report that MCTPs are plasmodesmata proteins that insert into the ER via their transmembrane region while their C2 domains dock to the PM through interaction with anionic phospholipids. A *Atmctp3/Atmctp4* loss of function mutant induces plant developmental defects, impaired plasmodesmata function and composition, while MCTP4 expression in a yeast Δtether mutant partially restores ER-PM tethering. Our data suggest that MCTPs are unique membrane tethers controlling both ER-PM contacts and cell-to-cell signalling.

**Keywords** ER-PM membrane contact sites; intercellular communication in plants; multiple C2 domains and transmembrane region proteins; plasmodesmata

**Subject Categories** Membrane & Intracellular Transport; Plant Biology

## Introduction

Intercellular communication is essential for the establishment of multicellularity, and evolution gave rise to distinct mechanisms to facilitate this process. Plants have developed singular intercellular pores—the plasmodesmata—which span the cell wall and interconnect nearly every single cell, establishing direct membrane and cytoplasmic continuity throughout the plant body [1]. Plasmodesmata are indispensable for plant life. They control the intercellular trafficking of non-cell-autonomous signals such as transcription factors, small RNAs, hormones and metabolites during key growth and developmental events [1–11]. Over the past few years, plasmodesmata have emerged as key components of plant defence signalling [12–14]. Mis-regulation of plasmodesmata function can lead to severe defects in organ growth and tissue patterning but also generate inappropriate responses to biotic and abiotic stresses [7,8,12,15–17]. Plasmodesmata not only serve as conduits, but act as

1 Laboratoire de Biogenèse Membranaire, UMR5200, CNRS, Université de Bordeaux, Villenave d'Ornon, France
2 Laboratoire de Biophysique Moléculaire aux Interfaces, TERRA Research Centre, GX ABT, Université de Liège, Gembloux, Belgium
3 Bordeaux Imaging Centre, Plant Imaging Platform, UMS 3420, INRA-CNRS-INSERM-University of Bordeaux, Villenave-d'Ornon, France
4 Department of Biosciences, University of Durham, Durham, UK
5 Division of Biochemistry, Department of Biology, Friedrich-Alexander University Erlangen-Nuremberg, Erlangen, Germany
6 Department of Protein Evolution, Max Planck Institute for Developmental Biology, Tübingen, Germany
7 Proteome Platform, Functional Genomic Center of Bordeaux, University of Bordeaux, Bordeaux Cedex, France
8 The Sainsbury Laboratory, University of Cambridge, Cambridge, UK
9 Biomedical Sciences Research Complex, University of St Andrews, Fife, UK
10 Cell and Molecular Sciences, The James Hutton Institute, Dundee, UK
  *Corresponding author. Tel: +44 1334 464829; E-mail: jt58@st-andrews.ac.uk
  **Corresponding author. Tel: +33 55712 2539; E-mail: emmanuelle.bayer@u-bordeaux.fr
  ¶These authors contributed equally to this work
  †Present address: Division of Biology and Biological Engineering, California Institute of Technology, Pasadena, CA, USA
  ‡Present address: UMR 1332 BFP, INRA, University of Bordeaux, Bordeaux, France
  §Present address: Matrice Extracellulaire et Dynamique Cellulaire MEDyC, UMR7369, CNRS, Université de Reims-Champagne-Ardenne, Reims, France

specialised signalling hubs, capable of generating and/or relaying signals from cell to cell through plasmodesmata-associated receptor activity [18–21].

Plasmodesmata are structurally unique [22,23]. They contain a strand of ER, continuous through the pores, tethered extremely tightly (~10 nm) to the PM by spoke-like elements [24,25] whose function and identity are unknown. Inside plasmodesmata, specialised subdomains of the ER and the PM co-exist, each being characterised by a unique set of lipids and proteins, both critical for proper function [6,12,26–31]. Where it enters the pores, the ER becomes constricted to a 15-nm tube (the desmotubule) leaving little room for lumenal trafficking. According to current models, transfer of molecules occurs in the cytoplasmic sleeve between the ER and the PM. Constriction of this gap, by the deposition of callose (β-1,3 glucan), in the cell wall around plasmodesmata, is assumed to be the main regulator of the pore size exclusion limit [4,32]. Recent work, however, suggests a more complex picture where the plasmodesmal ER-PM gap is not directly related to pore permeability and may play additional roles [22,25]. Newly formed plasmodesmata (type I) exhibit such close contact (~2–3 nm) between the PM and the ER that no electron-lucent cytoplasmic sleeve is observed [25]. During subsequent cell growth and differentiation, the pore widens, separating the two membranes, which remain connected by visible electron-dense spokes, leaving a cytosolic gap (type II). This transition has been proposed to be controlled by protein tethers acting at the ER-PM interface [22,33]. Counterintuitively, type I plasmodesmata with no apparent cytoplasmic sleeve are open to macromolecular trafficking and recent data indicate that tight ER-PM contacts may in fact favour transfer of molecules from cell to cell [25,34].

The close proximity of the PM and ER within the pores and the presence of tethers qualify plasmodesmata as a specialised type of ER-to-PM membrane contact site (MCS) [1,33]. MCS are structures found in all eukaryotic cells which function in direct interorganellar signalling by promoting fast, non-vesicular transfer of molecules and allowing collaborative action between the two membranes [35–46]. In yeast and mammalian cells, MCS protein tethers are known to physically bridge the two organelles, to control the intermembrane gap and to participate in organelle cross-talk. Their molecular identity/specificity dictate structural and functional singularity to different types of MCS [47,48]. To date, the plasmodesmal membrane tethers remain unidentified, but by analogy to other types of MCS, it seems likely that they play important roles in plasmodesmal structure and function, and given their unique position within a cell-to-cell junction may link intra- and intercellular communication.

Here, we have reduced the complexity of the previously published *Arabidopsis* plasmodesmal proteome [49] through the combination of a refined purification protocol [28,50,54] and semi-quantitative proteomics, to identify 115 proteins highly enriched in plasmodesmata and identify tether candidates. Amongst the most abundant plasmodesmal proteins, members of the multiple C2 domains and transmembrane region proteins (MCTPs) were enriched in post-cytokinetic plasmodesmata with tight ER-PM contact compared to mature plasmodesmata with wider cytoplasmic gap and sparse spokes, and exhibit the domain architecture characteristic of membrane tethers, with multiple lipid-binding C2 domains in the N-terminal and multiple transmembrane domains in the C-terminal region. Two MCTP members, AtMCTP1/Flower Locus T

Interacting Protein (FTIP) and AtMCTP15/QUIRKY (QKY), have previously been localised to plasmodesmata in *Arabidopsis* and are involved in cell-to-cell signalling [20,51]. However, two recent studies indicate that other MCTP members, including AtMCTP3, AtMCTP4 and AtMCTP9, which show high plasmodesmata enrichment in our proteome, do not associate with the pores *in vivo* [52,53]. Using confocal live cell imaging, 3D structured illumination super-resolution microscopy, correlative light and electron microscopy, immunogold labelling and genetic approaches, we provide evidence that MCTPs, including AtMCTP3 and 4, localise and function at plasmodesmata pores. We further show that *Atmctp3/Atmctp4* loss of function *Arabidopsis* mutant, which displays developmental phenotypic defects, shows reduced cell-to-cell trafficking and a significantly altered plasmodesmata proteome. By combining confocal imaging of truncated MCTP mutants, molecular dynamics and yeast complementation, our data indicate that MCTP properties are consistent with a role in ER-PM membrane tethering at plasmodesmata. As several MCTP members have been identified as important components of plant intercellular signalling [20,51], our data suggest a link between interorganelle contacts at plasmodesmata and intercellular communication in plants.

## Results

### Identification of plasmodesmal ER-PM tethering candidates

To identify putative plasmodesmal MCS tethers, we decided to screen the plasmodesmata proteome for ER-associated proteins (a general trait of ER-PM tethers [47,48]) with structural features enabling bridging across two membranes. A previously published plasmodesmata proteome reported the identification of more than 1,400 proteins in *Arabidopsis* [49], making the discrimination of true plasmodesmata-associated from contaminant proteins a major challenge. To reduce the proteome complexity and identify core plasmodesmata protein candidates, we used a refined plasmodesmata purification technique [28,50,54] together with label-free comparative quantification (Appendix Fig S1A). Plasmodesmata and likely contaminant fractions, namely the PM, microsomal, total cell and cell wall fractions, were purified from 6-day-old *Arabidopsis* suspension culture cells and simultaneously analysed by liquid chromatography–tandem mass spectrometry (LC-MS/MS). For each protein identified, its relative enrichment in the plasmodesmata fraction versus "contaminant" fractions was determined (Appendix Fig S1B; Appendix Table S1). Enrichment ratios for selecting plasmodesmal candidates were set based on previously characterised plasmodesmal proteins (see Materials and Methods for details). This refined proteome dataset was reduced to 115 unique proteins, cross-referenced with two published ER proteomes [55,56] and used as a basis for selecting MCS-relevant candidates.

Alongside, we also analysed changes in protein abundance during the ER-PM tethering transition from very tight contacts in post-cytokinetic plasmodesmata (type I) to larger ER-PM gap and sparse tethers in mature plasmodesmata (type II) [25]. For this, we obtained a similar semi-quantitative proteome from 4- and 7-day-old culture cells, enabling a comparison of plasmodesmata composition during the tethering transition [25] (Appendix Fig S2).

A survey of our refined proteome identified several members of the multiple C2 domains and transmembrane region proteins (MCTPs) family, namely AtMCTP3-7, AtMCTP9, AtMCTP10 and 14–16, as both abundant and highly enriched at plasmodesmata (Appendix Fig S1B, Appendix Table S1). In addition to being plasmodesmata-enriched proteins, our data also suggest that MCTPs are differentially regulated during the ER-PM tethering transition from post-cytokinetic to mature plasmodesmata [25] (Appendix Fig S2). Amongst the 47 plasmodesmal proteins differentially enriched, all MCTPs were more abundant (1.4–3.6 times) in type I (tight ER-PM contacts) compared with type II (open cytoplasmic sleeves) plasmodesmata (Appendix Fig S2).

**MCTPs are ER-associated proteins located at plasmodesmata and present structural features of membrane tethers**

MCTPs are structurally reminiscent of the ER-PM tether families of mammalian extended-synaptotagmins (HsE-Syts) and *Arabidopsis* synaptotagmins (AtSYTs) [57,58], possessing lipid-binding C2 domains at one end and multiple transmembrane domains (TMDs) at the other, a domain organisation consistent with the function of membrane tethers (Appendix Fig S3). Unlike HsE-Syts and AtSYTs, the transmembrane region of MCTPs is located at the C-terminus and three to four C2 domains at the N-terminus (Fig 1A; Appendix Fig S3). Two members of the *Arabidopsis* MCTP family, AtMCTP1/Flower Locus T Interacting Protein (FTIP) and AtMCTP15/QUIRKY (QKY), have previously been localised to plasmodesmata in *Arabidopsis* and implicated in cell-to-cell trafficking of developmental signals [20,51]. However, two recent studies indicate that other MCTP members, including AtMCTP3, AtMCTP4 and AtMCTP9, which show high plasmodesmata enrichment in our proteome, do not associate with the pores *in vivo* [52,53].

We investigated the *in vivo* localisation of MCTPs identified in our proteomic screen by transiently expressing N-terminal fusions with fluorescent proteins in *Nicotiana benthamiana* leaves. As the MCTP family is conserved in *N. benthamiana* (Appendix Fig S4) and to avoid working in a heterologous system, we also examined the localisation of NbMCTP7, whose closest homolog in *Arabidopsis* was also identified as highly enriched in plasmodesmata fractions (AtMCTP7; Appendix Fig S1). Confocal imaging showed that all selected MCTPs, namely AtMCTP3, AtMCTP4, AtMCTP6, AtMCTP9 and NbMCTP7, displayed a similar subcellular localisation, with a faint ER-like network at the cell surface and a punctate distribution along the cell periphery at sites of epidermal cell-to-cell contacts (Fig 1B and C). Time-lapse imaging showed that peripheral fluorescent punctae were immobile, which contrasted with the high mobility of the ER-like network (Movie EV1). Co-localisation with RFP-HDEL confirmed MCTPs association with the cortical ER, while the immobile spots at the cell periphery perfectly co-localised with the plasmodesmal marker mCherry-PDCB1 ([27,29]; Fig 1C). Co-labelling with general ER-PM tethers such as VAP27.1-RFP and SYT1-RFP [57,59] showed partial overlap with GFP-NbMCTP7, while co-localisation with mCherry-PDCB1 was significantly higher (Appendix Fig S5). To further quantify and ascertain MCTP association with plasmodesmata, we measured a plasmodesmal enrichment ratio, hereafter named "plasmodesmata index". For this, we calculated fluorescence intensity at plasmodesmata pit fields (indicated by mCherry-PDCB1 or aniline blue) versus cell periphery. All MCTPs tested displayed a high plasmodesmata index, ranging from 1.85 to 4.15, similar to PDLP1 (1.36) and PDCB1 (1.45), two well-established plasmodesmata markers [29,60] (Fig 1D), confirming enrichment of MCTPs at pit fields. When stably expressed in *Arabidopsis thaliana* under the moderate promoter UBIQUITIN10 or 35S promoter AtMCTP3, AtMCTP4, AtMCTP6 and AtMCTP9 were found mainly restricted to plasmodesmata (Appendix Fig S6A, white arrows), as indicated by an increase in their plasmodesmata index compared with transient expression in *N. benthamiana* (Appendix Fig S6B). A similar increase in the plasmodesmata index is seen with PDLP1.RFP when stably expressed in *Arabidopsis* (Appendix Fig S6B). A weak but consistent ER localisation was also visible in stably transformed *Arabidopsis* (Appendix Fig S6A red stars).

To get a better understanding of MCTP distribution within the plasmodesmal pores, we further analysed transiently expressed GFP-NbMCTP7 by 3D structured illumination super-resolution microscopy (3D-SIM) [61] (Fig 1E). We found that NbMCTP7 is associated with all parts of plasmodesmata including the neck regions and central cavity, as well as showing continuous fluorescence throughout the pores. In some cases, lateral branching of plasmodesmata within the central cavity was resolved. The very faint continuous fluorescent threads connecting neck regions and central cavity correspond to the narrowest regions of the pores and may indicate association with the central desmotubule (Fig 1E, white arrows).

**Figure 1. MCTPs are ER-associated proteins located at plasmodesmata.**
Localisation of AtMCTP3, AtMCTP4, AtMCTP6, AtMCTP9 and NbMCTP7 in *N. benthamiana* epidermal cells visualised by confocal microscopy. MCTPs were tagged at their N-terminus with YFP or GFP and expressed transiently under 35S (NbMCTP7) or UBIQUITIN10 promoters (AtMCTP3, AtMCTP4, AtMCTP6 and AtMCTP9).

A Schematic representation of MCTP domain organisation, with three to four C2 domains at the N-terminus and multiple transmembrane domains (TMD) at the C-terminus.

B GFP-NbMCTP7 associates with punctae at the cell periphery (white arrowheads) and labels a reticulated network at the cell surface resembling the cortical ER. Maximum projection of z-stack. Scale bar, 2 μm.

C Single optical sections at cell surface (left) or cell-to-cell interface (right), showing the co-localisation between MCTPs and the ER marker RFP-HDEL (left) and the plasmodesmata marker mCherry-PDCB1 (right). Intensity plots along the white dashed lines are shown for each co-localisation pattern. Scale bars, 2 μm.

D The plasmodesmata (PD) index of individual MCTPs is above 1 (red dashed line) and similar to known plasmodesmata markers (aniline blue, PDCB1, PDLP1) confirming enrichment at plasmodesmata. By comparison, the PM-localised proton pump ATPase PMA2 and the ER marker HDEL that are not enriched at plasmodesmata have a PD index below 1. In the box plot, median is represented by horizontal line, values between quartiles 1 and 3 are represented by box ranges, and minimum and maximum values are represented by error bars. Three biological replicates were analysed.

E 3D-SIM images (individual z-sections) of GFP-NbMCTP7 within three different pit fields (panels 1-2, 3-4 and 5, respectively) showing fluorescence signal continuity throughout the pores, enrichment at plasmodesmal neck regions (1-2, arrowheads in 1), central cavity (3-4, arrowhead in 3) and branching at central cavity (5, arrow). Dashed lines indicate position of cell wall borders. Scale bars, 500 nm.

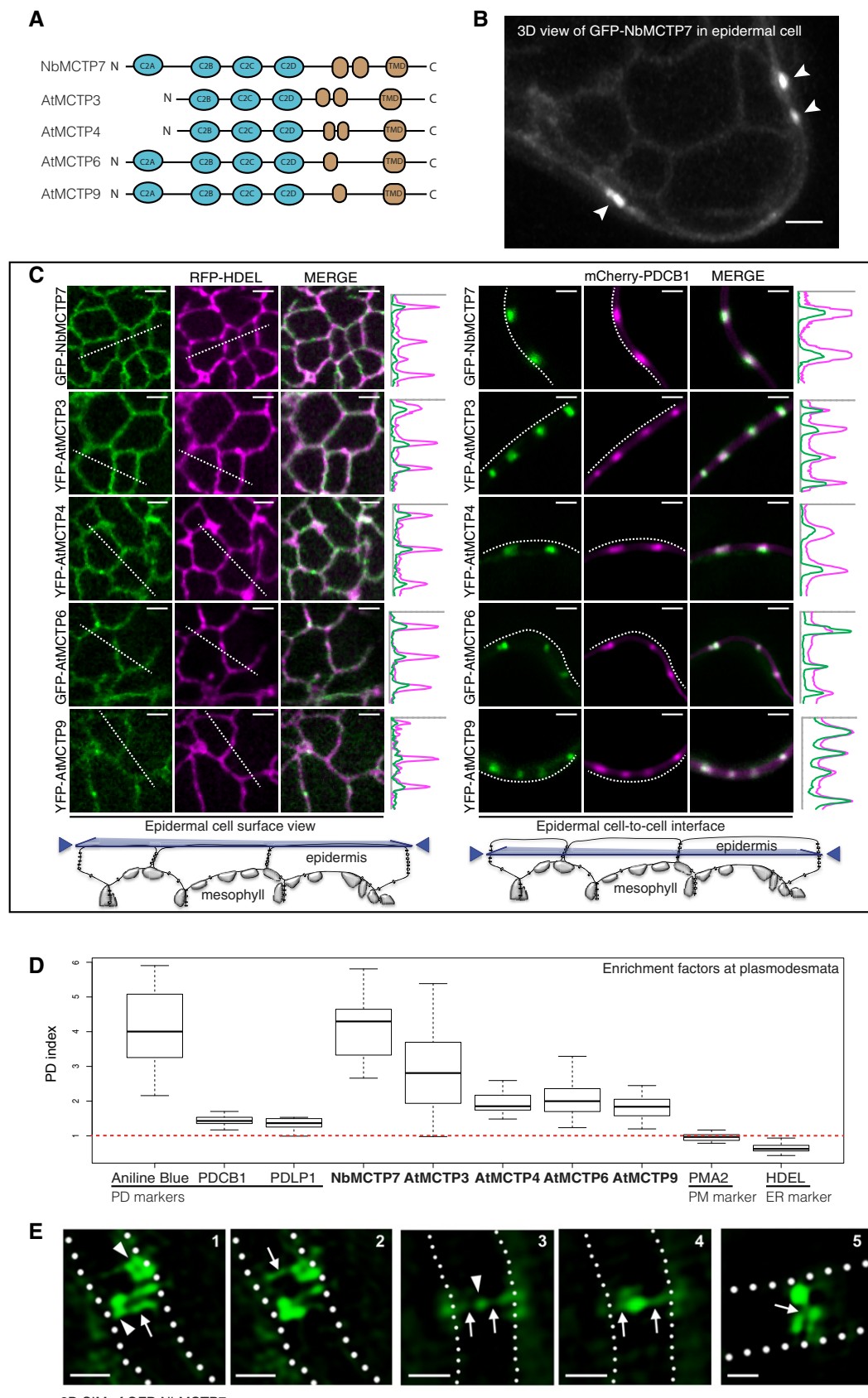

Figure 1.

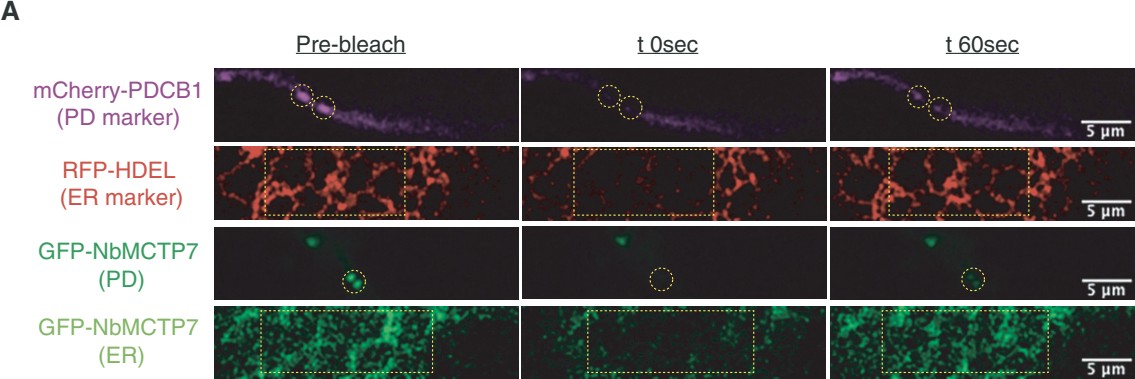

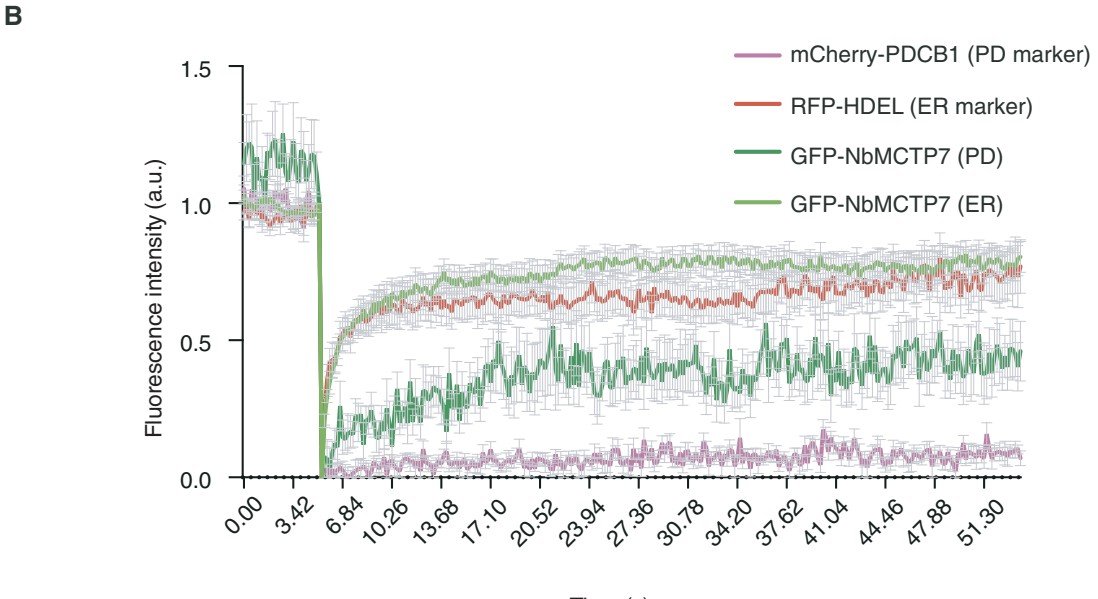

**Figure 2. NbMCTP7 mobility at plasmodesmata is reduced compared to cortical ER. FRAP analysis of NbMCTP7 in *N. benthamiana* leaf epidermal cells.**

A   Representative pre-bleach and post-bleach images for mCherry-PDCB1 (purple; plasmodesmata marker), RFP-HDEL (red; ER marker) and GFP-NbMCTP7 at plasmodesmata (dark green) and at the cortical ER (light green). Yellow dashed boxes or circles indicate the bleach region.
B   FRAP comparing the mobility of GFP-NbMCTP7 at plasmodesmata (dark green) and at the cortical ER (light green) to that of RFP-HDEL (red) and mCherry-PDCB1 (purple). NbMCTP7 is highly mobile when associated with the ER as indicated by fast fluorescent recovery but shows reduced mobility when associated with plasmodesmata. Data are averages of at least 3 separate experiments; error bars indicate standard error.

Using fluorescence recovery after photobleaching (FRAP), we then assessed the mobility of NbMCTP7. We found that, when associated with the cortical ER, the fluorescence recovery rate of GFP-NbMCTP7 was extremely fast and similar to RFP-HDEL with half-times of 1.16 and 0.99 s, respectively (Fig 2A and B). By contrast, when GFP-NbMCTP7 was associated with plasmodesmata, the recovery rate slowed down to a half-time of 4.09 s, indicating restricted mobility, though still slightly faster than for the cell wall-localised plasmodesmal marker mCherry-PDCB1 (5.98 s). Overall, these results show that NbMCTP7 mobility is high at the cortical ER but becomes restricted inside the pores.

From our data, we concluded that MCTPs are ER-associated proteins, whose members specifically and stably associate with plasmodesmata. They display the structural features required for ER-PM

tethering and are differentially associated with the pores during the transition in ER-PM contacts.

## Loss of function *mctp3/mctp4* double mutant shows pleiotropic developmental defects, reduced cell-to-cell trafficking and an altered plasmodesmata proteome

We next focused on AtMCTP4, which according to our proteomic screen appears as one of the most abundant proteins associated with plasmodesmata-enriched fractions (Appendix Table S1). The implication of AtMCTP4 association with plasmodesmata is that the protein contributes functionally to cell-to-cell signalling. Given the importance of plasmodesmata in tissue patterning and organ growth, a loss-of-function mutant is expected to show defects in

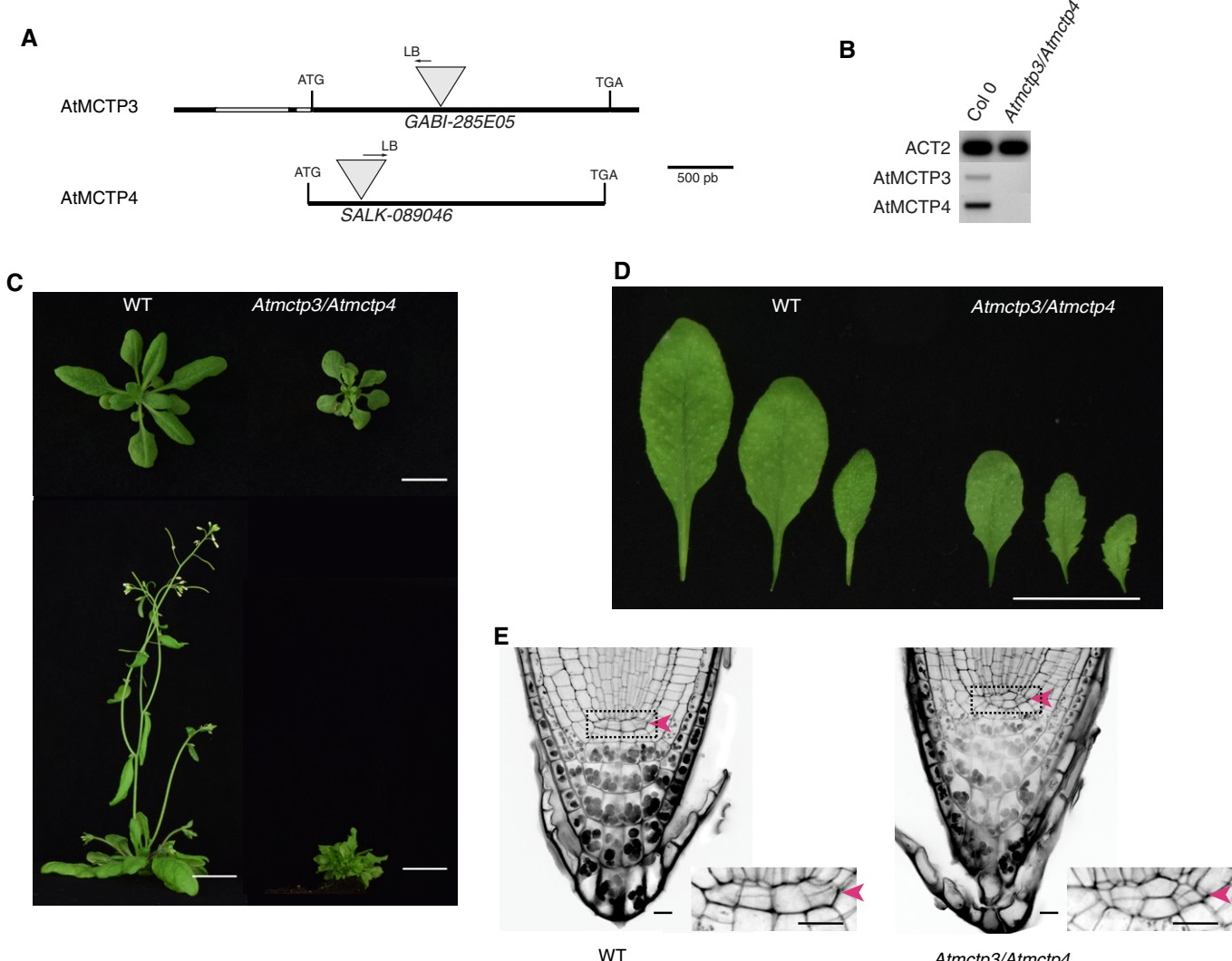

**Figure 3. *Atmctp3/Atmctp4* loss of function double mutant shows severe defects in development.**

A–E Characterisation of *Atmctp3/Atmctp4* double mutant in *Arabidopsis*. (A) Schematic representation of T-DNA insertions in *AtMCTP3* and *AtMCTP4*. LB, left border. (B) RT–PCR analysis of AtMCTP3, AtMCTP4 and Actin2 (ACT2) transcripts in Col-0 wild-type (WT) and *Atmctp3/Atmctp4* double mutant showing the absence of full-length transcripts in the *Atmctp3/Atmctp4* double mutant. (C) Rosette and inflorescence stage phenotypes of *Atmctp3/Atmctp4* double mutant compared to Col-0 WT. Scale bar, 2 cm. (D) Leaf phenotypes of *Atmctp3/Atmctp4* double mutant compared to WT. Scale bar, 2 cm. (E) Pseudo-Schiff propidium iodide method-stained root tips of WT and *Atmctp3/Atmctp4* double mutant. Defect in quiescent centre (QC, red arrowheads) cell organisation was observed in 20 out of 20 plants examined. Scale bars, 10 μm.

plant development. We first obtained T-DNA insertion lines for AtMCTP4 and its closest homolog AtMCTP3, which share 92.8% identity and 98.7% similarity in amino acids with AtMCTP4, but both single knockouts showed no apparent phenotypic defects (Appendix Fig S7). We therefore generated an *Atmctp3/Atmctp4* double mutant, which presented pleiotropic developmental defects with a severely dwarfed and bushy phenotype, twisted leaves with increased serration (Fig 3A–D) and multiple inflorescences (Appendix Fig S7). The phenotype was fully complemented by YFP-AtMCTP3 expression (Appendix Fig S7). While preparing this manuscript, another paper describing the *Atmctp3/Atmctp4* mutant was published [50], reporting similar developmental defects. We noted additional phenotypic defects in particular aberrant

patterning in the root apical meristem, specifically within the quiescent centre (QC) and columella cells (Fig 3E, Appendix Fig S14). Instead of presenting the typical four-cell layer organisation, we observed asymmetrical divisions in the QC of the *Atmctp3/Atmctp4*, suggesting that both proteins may play a general role in cell stem niche maintenance [50]. To further investigate the role of AtMCTP3 and AtMCTP4 in plasmodesmata function, we performed intercellular trafficking assays by monitoring GFP-sporamin (47 kDa) [62] movement from single-cell transformation sites in fully expanded leaves. Compared to wild-type Col-0 or *Atmctp3* and *Atmctp4* single mutants, *Atmctp3/Atmctp4* double mutant presented a significant reduction of GFP-sporamin spread, indicating reduced plasmodesmata-mediated macromolecular

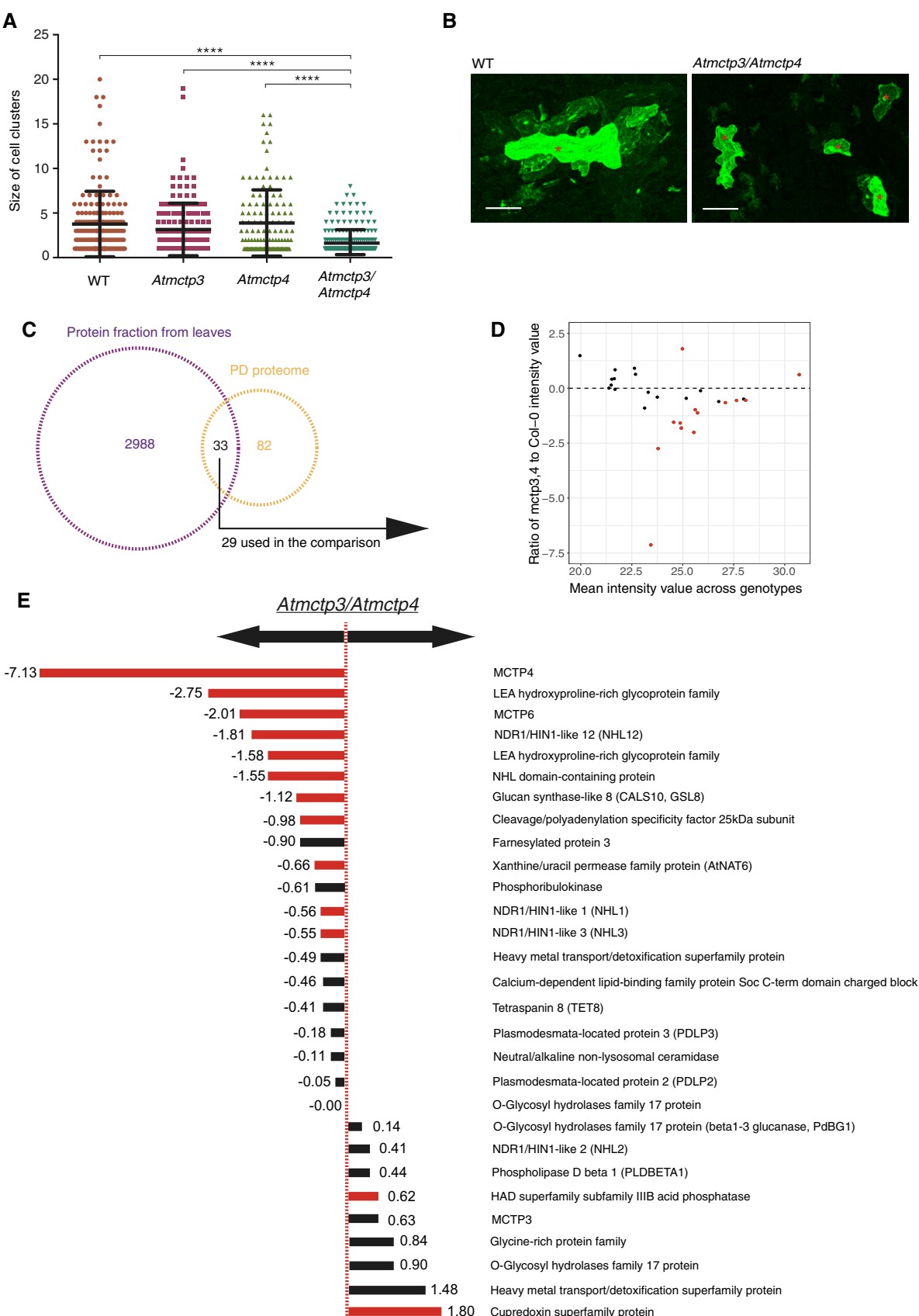

Figure 4.

**Figure 4.  Plasmodesmata function and composition are altered in *Atmctp3/Atmctp4 Arabidopsis* mutant.**

A, B  Macromolecular trafficking through plasmodesmata is reduced in *Atmctp3/Atmctp4 Arabidopsis* mutant. Leaves of Col-0, *Atmctp3*, *Atmctp4* single mutants and *Atmctp3/Atmctp4* double mutant were bombarded with a GFP-sporamin (47 kDa) expression plasmid. Diffusion of GFP-sporamin to surrounding cells 72 h after bombardment was used as a measure of plasmodesmata molecular trafficking. (A) Scatter plot representation with the black lines representing the mean value (middle) and SD. (B) Representative GFP-sporamin fluorescent foci observed by confocal microscopy 72 h after bombardment. Bombarded cells are indicated by a red star. *n* = 50 foci for Col-0; *n* = 37 foci for *Atmctp3* single mutant; *n* = 33 foci for *Atmctp4* single mutant; and *n* = 48 foci for *Atmctp3/Atmctp4* double mutant (3 biological replicates). Pairwise comparisons by Wilcoxon test, ****$P < 0.0001$. Scale bar, 100 μm.

C–E  Comparative proteomic analysis of *Atmctp3/Atmctp4* and Col-0. (C) Intersection between the protein identities obtained from leaf tissue and those in the plasmodesmata proteome (Appendix Table S1). Only proteins appearing in at least 3/5 samples were employed for the comparison (29). (D) Plot of the log2 ratios of the average plasmodesmata protein intensities in *Atmctp3/Atmctp4* relative to Col-0. Dashed line indicates no change (0 = log2(1)), above the line enrichments, below the line depletions. (E) List of plasmodesmata proteins detected in leaf tissue and their ratios of abundance in *Atmctp3/Atmctp4* relative to Col-0. Please note that we detected AtMCTP3 unique peptides at relatively high abundance in the *Atmctp3/Atmctp4* mutant. These peptides are located before the T-DNA insertion, and it is therefore likely that a truncated non-functional protein is still translated at levels similar to the wild-type (see Appendix Fig S8). Bars for the proteins whose differential abundance is supported by statistical testing (*t*-test with *P*-value correction for false discovery rate. $P < 0.05$) are in red. Bars for other proteins are in black.

trafficking (Fig 4A and B). To complement the trafficking assays, we performed comparative proteomic profiling of the *Atmctp3/Atmctp4* mutant and wild-type Col-0. For this, we analysed cell wall extracts from fully expanded leaves using quantitative high-resolution mass spectrometry [63]. Comparative data analysis showed that about one-third (13 out of 29) of the proteins identified as plasmodesmata-associated according to our "refined" proteome were differentially regulated in *Atmctp3/Atmctp4* mutant compared to wild-type Col-0 (Fig 4C–E). This indicates that the molecular composition of plasmodesmata is substantially altered in the *Atmctp3/Atmctp4* mutant. Altogether, these data indicate that *Atmctp3/Atmctp4* loss of function is detrimental for plasmodesmata function and composition, and this probably contributes to developmental defects observed in the mutant.

## AtMCTP4 is a plasmodesmata-associated protein

To further verify plasmodesmata association of AtMCTP4, we expressed the N-terminal GFP fusion under its own promoter (pAtMCTP4:GFP-AtMCTP4). At the tissue level, the expression pattern of this construct in stable *Arabidopsis* lines was consistent with the phenotypic defects we observed in the *Atmctp3/Atmctp4* mutant, as strong expression was observed in the inflorescence shoot apical meristem (Fig 5A), root tip (including QC), lateral root primordia and young leaf primordia (Appendix Fig S9A), i.e., the tissues showing developmental defects in the mutant (Fig 3; [50]). However, AtMCTP4 has recently been reported as an endosomal-localised protein [52], which is in conflict with our data indicating plasmodesmata association. We analysed localisation pattern of pAtMCTP4:GFP-AtMCTP4 at the subcellular level in *Arabidopsis* stable lines. Similar to transient expression experiments (Fig 1), we found that pAtMCTP4:GFP-AtMCTP4 was located at stable punctate spots at the cell periphery (Fig 5A white arrows; Movie EV2), in all tissues examined, *i.e.* leaf epidermal and spongy mesophyll cells, hypocotyl epidermis, lateral root primordia, root tip and inflorescence shoot apical meristem. These immobile dots co-localised perfectly with aniline blue indicating plasmodesmata association (Fig 5A top row), which was also evident in leaf spongy mesophyll cells where the dotty pattern of pAtMCTP4:GFP-AtMCTP4 was present on adjoining walls (containing plasmodesmata), but absent from non-adjoining walls (without plasmodesmata) (Fig 5A white arrowheads). Furthermore, localisation of pAtMCTP4:GFP-AtMCTP4 at interfaces between epidermal pavement cells and stomata guard

cells, where plasmodesmata are only half-formed on the pavement cell side [64], is similar to that of the viral movement protein cucumber mosaic virus (CMV 3a) (Appendix Fig S9B). We also observed a weak but consistent ER association of AtMCTP4 (Fig 5A, red stars).

To investigate further AtMCTP4 association with plasmodesmata, we performed immunogold labelling on high-pressure frozen freeze-substituted root sections of 6-day-old pAtMCTP4:GFP-AtMCTP4 seedlings, using anti-GFP antibodies. GFP-AtMCTP4-associated gold particle signal was seen along the length of the channel and neck region (Fig 5B). Distribution of the gold labelling showed a strong preference for labelling associated with plasmodesmata. Immunogold labelling of wild-type roots gave no significant labelling (Appendix Fig S11A). To complement immunogold labelling on root sections, we also performed correlative light and electron microscopy (CLEM) on walls purified from pAtMCTP4:GFP-AtMCTP4 seedlings. Calcofluor staining combined with confocal imaging revealed discrete plasmodesmata pit fields characterised by the absence of cellulose staining [65] into which GFP-AtMCTP4 punctate signal was systematically associated (Fig 5C, Appendix Fig S10, yellow arrowheads). We then transferred the wall fragments to electron microscopy, after negative staining with uranyl acetate, to reveal plasmodesmata structures. As shown in Fig 5D and Appendix Fig S11B, the dotty GFP-AtMCTP4 signal was perfectly co-localised with plasmodesmata pit fields. To further confirm the results from the CLEM, we performed immunogold labelling using anti-callose (10 nm gold) and/or anti-PDCB (5 nm gold), two well-established plasmodesmata markers [27,29]. Both antibodies specifically labelled the areas identified as pit fields (Fig 5D, Appendix Fig S11C). In summary, we concluded that whatever the tissue and organ considered, AtMCTP4 is strongly and consistently associated with plasmodesmata but also presents a steady association with the ER.

## The C-terminal transmembrane regions of MCTPs serve as ER anchors

A requirement for tethers is that they physically bridge two membranes. Often this is achieved through lipid-binding module(s) at one terminus of the protein and transmembrane domain(s) at the other [47,48]. All sixteen *Arabidopsis* MCTPs contain two to three predicted TMDs near their C-terminus (collectively referred to as the transmembrane region, TMR). To test whether the MCTP TMRs are

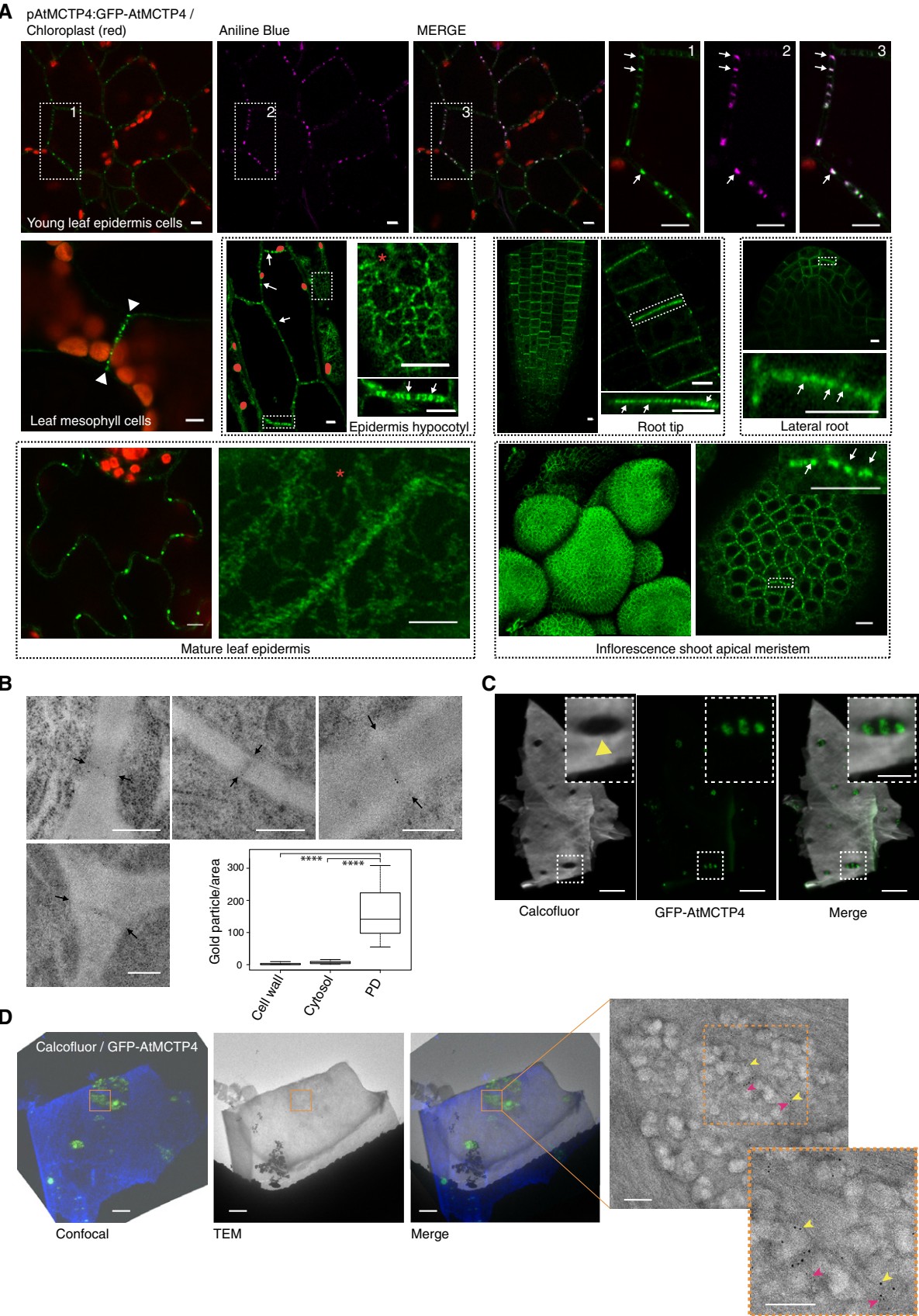

**Figure 5.**

**Figure 5. AtMCTP4 is a plasmodesmata protein localised at plasmodesmata pit fields.**

A   Subcellular localisation of GFP-AtMCTP4 under AtMCTP4 native promoter in *Arabidopsis* transgenic lines visualised by confocal microscopy. In all tissues examined, GFP-MCTP4 shows a typical punctate distribution of plasmodesmata at the cell boundaries (indicated by white arrows). In leaf, spongy mesophyll GFP-AtMCTP4 punctate pattern was visible only on adjoining walls (arrowheads), which contain plasmodesmata but absent from non-adjoining walls. GFP-AtMCTP4 dots at the cell periphery are immobile (see Movie EV2) and co-localise perfectly with aniline blue (top row) confirming plasmodesmata localisation. In most tissues examined, an ER-reticulated pattern was also observable (red stars). Boxed regions are magnified in adjacent panels. Please note that the hypocotyl epidermis was imaged in airy scan mode and chloroplasts were manually outlined in red. Scale bars, 5 µm.

B   Immunogold localisation of GFP-AtMCTP4 to plasmodesmata. Thin sections of pAtMCTP4:GFP-AtMCTP4 transgenic *Arabidopsis* roots subjected to immunogold labelling using anti-GFP antibodies (5 and 10 nm gold particles). Electron micrographs showing wall sections with plasmodesmata labelled by gold particles. Gold particles were quantified relative to the area occupied by plasmodesmata, cell wall and cytosol, respectively (see Materials and Methods). ****$P < 0.001$ in pairwise Wilcoxon test (total particles count 298). In the box plot, median is represented by horizontal line, values between quartiles 1 and 3 are represented by box ranges, and minimum and maximum values are represented by error bars. Scale bars, 300 nm

C   Confocal observation of cell walls purified from pAtMCTP4:GFP-AtMCTP4. Cell wall was stained with calcofluor, revealing plasmodesmata pit fields where calcofluor staining, hence cellulose, is absent/reduced (yellow arrowhead). GFP-AtMCTP4 signal is systematically associated with plasmodesmata pit fields (see boxed magnified region and Appendix Fig S10). Scale bars, 5 and 2.5 µm in boxed regions.

D   CLEM on cell walls purified from pAtMCTP4:GFP-AtMCTP4 *Arabidopsis* seedlings combined with immunogold labelling against callose (yellow arrow; 10-nm gold particles) and PDCB1 protein (magenta arrow; 5-nm gold particles), two plasmodesmata markers. TEM = transmission electron microscopy. For CLEM, also see Appendix Fig S11. Scale bars, 5 µm for confocal images and 200 nm for TEM images.

determinants of ER insertion, we generated truncation mutants lacking the C2 domains for NbMCTP7, AtMCTP3, AtMCTP4, AtMCTP6, AtMCTP9 as well as AtMCTP1/FTIP and AtMCTP15/QKY (Fig 6A). When fused to YFP at their N-terminus, all truncated mutants retained ER association, as demonstrated by co-localisation with RFP-HDEL (Fig 6B left panels). Meanwhile, plasmodesmata association was completely lost and the plasmodesmata index of all truncated MCTP_TMRs dropped below one, comparable to RFP-HDEL (Fig 6B right panels and c), quantitatively confirming the loss of plasmodesmata association when the C2 modules were deleted. For AtMCTP15/QKY, this is in agreement with a previous study [20]. We therefore concluded that, similar to the HsE-Syt and AtSYT ER-PM tether families [57,58,66], MCTPs insert into the ER through their TMRs, but the TMR alone is not sufficient for MCTP plasmodesmal localisation.

## MCTP C2 domains can bind membranes in an anionic lipid-dependent manner

Members of the HsE-Syt and AtSYT tether families bridge across the intermembrane gap and dock to the PM via their C2 domains [57,58,67,68]. *Arabidopsis* MCTPs contain three to four C2 domains, which may also drive PM association through interactions with membrane lipids. C2 domains are independently folded structural and functional modules with diverse modes of action, including membrane docking, protein–protein interactions and calcium sensing [69].

To investigate the function of MCTP C2 modules, we first searched for homologs of AtMCTP individual C2 domains (A, B, C, and D) amongst all human and *A. thaliana* proteins using the HHpred web server [70] for remote homology detection. The searches yielded a total of 1,790 sequence matches, which contained almost all human and *A. thaliana* C2 domains. We next clustered the obtained sequences based on their all-against-all pairwise similarities in CLANS [71]. In the resulting map (Appendix Fig S12A), the C2 domains of *Arabidopsis* MCTPs (AtMCTPs, coloured cyan) most closely match the C2 domains of membrane-trafficking and membrane-tethering proteins, including human MCTPs (HsMCTPs, green), human synaptotagmins (HsSyts, orange), human Ferlins (HsFerlins, blue), human HsE-Syts (HsE-Syts, magenta) and *Arabidopsis* SYTs (AtSYTs, red), most of which dock to membranes through direct interaction with anionic lipids [58,68,72–74]. By

comparison to the C2 domains of these membrane-trafficking and membrane-tethering proteins, the C2 domains of most other proteins do not make any connections to the C2 domains of AtMCTPs at the *P*-value cut-off chosen for clustering (1e-10). Thus, based on sequence similarity, the plant AtMCTP C2 domains are expected to bind membranes.

We next asked whether the C2 modules of MCTPs are sufficient for PM association *in vivo*. Fluorescent protein fusions of the C2A-D or C2B-D modules without the TMR were generated for NbMCTP7, AtMCTP3, AtMCTP4, AtMCTP6, AtMCTP9 as well as AtMCTP1/FTIP and AtMCTP15/QKY and expressed in *N. benthamiana*. We observed a wide range of subcellular localisations from cytosolic to PM-associated and in all cases plasmodesmata association was lost (Appendix Fig S12B–D).

To further investigate the potential for MCTP C2 domains to interact with membranes, we employed molecular dynamic modelling. We focussed on AtMCTP4, as a major plasmodesmal constituent and whose loss of function in conjunction with AtMCTP3 induces severe plant developmental and plasmodesmata defects [52] (Fig 3). We first generated the 3D structures of all three C2 domains of AtMCTP4 using 3D homology modelling and then tested the capacity of individual C2 domains to dock to membrane bilayers using coarse-grained dynamic simulations (Fig 7A; Movie EV3). Molecular dynamic modelling was performed on three different membranes: (i) a neutral membrane composed of phosphatidylcholine (PC), (ii) a membrane with higher negative charge composed of PC and phosphatidylserine (PS; 3:1) and (iii) a PM-mimicking lipid bilayer, containing PC, PS, sitosterol and the anionic phosphoinositide phosphatidyl inositol-4-phosphate (PI4P; 57:19:20:4). The simulations showed that all individual C2 domains of AtMCTP4 can interact with lipids and dock on the membrane surface when a "PM-like" lipid composition was used (Fig 7A). The PC-only membrane showed only weak interactions, while the PC:PS membrane allowed only partial docking (Fig 7A). Docking of AtMCTP4 C2 domains arose mainly through electrostatic interactions between lipid polar heads and basic amino acid residues at the protein surface. We also confirmed membrane docking and stable anionic lipid interaction for individual AtMCTP4 C2 domains using all atom simulation. For that, coarse-grained systems were transformed back to all atom representations and simulation was run for 100 ns to check the stability of membrane

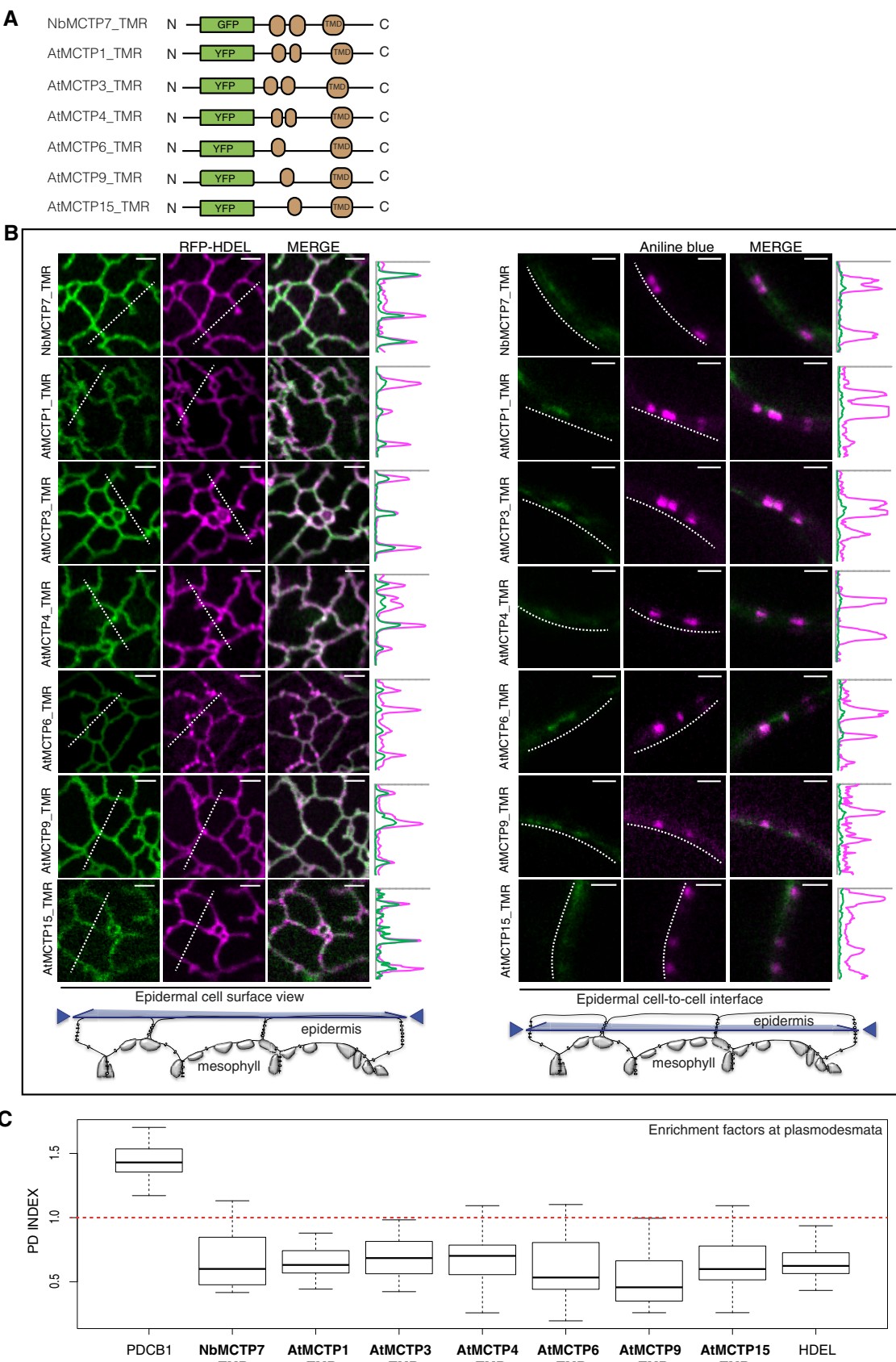

**Figure 6.**

◄

**Figure 6.  MCTPs insert into the ER membrane via their C-terminal transmembrane region.**

Localisation of truncated AtMCTP1, AtMCTP3, AtMCTP4, AtMCTP6, AtMCTP9, AtMCTP15 and NbMCTP7 transmembrane regions (TMR) in *N. benthamiana* leaf epidermal cells. TMRs were tagged at their N-terminus with GFP/YFP and expressed transiently under moderate UBIQUITIN10 promoter.

A   Schematic representation of truncated MCTPs tagged with GFP/YFP.

B   Optical sections at cell surface (left) and cell-to-cell interface (right) showing the co-localisation between GFP/YFP-MCTP_TMR constructs and the ER marker RFP-HDEL (left) and the plasmodesmata marker aniline blue (right). Intensity plots along the white dashed lines are shown for each co-localisation pattern. When expressed in epidermal cells, GFP/YFP-MCTP_TMR constructs associate with the ER but plasmodesmata association is lost. Scale bars, 2 μm.

C   The PD index of individual truncated MCTP_TMR constructs is below 1 (red dashed line), similar to the ER marker RFP-HDEL confirming loss of plasmodesmata localisation. In the box plot, median is represented by horizontal line, values between quartiles 1 and 3 are represented by box ranges, and minimum and maximum values are represented by error bars. Three biological replicates were analysed.

docking (Appendix Fig S13C). We further tested two other MCTP members, namely AtMCTP15/QKY and NbMCTP7, which possess four rather than three C2 domains. We found that similar to AtMCTP4, the individual C2 domains of AtMCTP15/QKY and NbMCTP7 exhibited membrane interaction in the presence of the negatively charged lipids (Appendix Fig S13A and B).

Our molecular dynamic data thus suggest that membrane docking of the AtMCTP4 C2 domains depends on the electrostatic charge of the membrane and more specifically on the presence of PI4P, a negatively charged lipid which has been reported as controlling the electrostatic field of the PM in plants [75].

To confirm the importance of PI4P for MCTP membrane interactions and, thus, potentially subcellular localisation, we used a short-term treatment with phenylarsine oxide (PAO), an inhibitor of PI4-kinases [75]. We focused on *Arabidopsis* root tips where effects of PAO have been thoroughly characterised [75]. In control-treated roots of *Arabidopsis* plants stably expressing UB10: YFP-AtMCTP4, the fluorescent signal was most prominent at the apical–basal division plane of epidermal root cells, where numerous plasmodesmata are established during cytokinesis [27] (Fig 7B white arrowheads). The YFP-AtMCTP4 fluorescence pattern was punctate at the cell periphery, each spot of fluorescence corresponding to a single or group of plasmodesmata (Fig 7C, white arrows). We found that 40-min treatment with PAO (60 μM) induced a loss in the typical spotty plasmodesmata-associated pattern, and instead, AtMCTP4 became more homogenously distributed along the cell periphery (Fig 7B and C). To confirm the effect of PAO on the cellular PI4P pool, we used a PI4P biosensor (1×PH FAPP1) which showed a clear shift from PM association to cytosolic localisation upon treatment with PAO [73] (Fig 7B). This control not only demonstrates that the PAO treatment was successful, but also highlights that the majority of PI4P was normally

found at the PM, rather than the ER, of *Arabidopsis* root cells. Therefore, the effect of PAO on YFP-AtMCTP4 localisation is likely related to a perturbance of PM docking by the MCTP4 C2 domains. When *Arabidopsis* seedlings were grown on PAO (1 and 10 μM) for 7 days, we observed root tip phenotypic defects reminiscent of the *Atmctp3/Atmctp4* mutant (Appendix Fig S14).

Altogether, our data suggest that the C2 domains of plant MCTPs can dock to membranes in the presence of negatively charged phospholipids and that PI4P depletion reduced AtMCTP4 stable association with plasmodesmata.

### AtMCTP4 expression is sufficient to partially restore ER-PM contacts in yeast

To further test the ability of MCTPs to physically bridge across membranes and tether the ER to the PM, we used a yeast Δtether mutant line deleted in six ER-PM tethering proteins resulting in the separation of the cortical ER (cER) from the PM [76] and expressed untagged AtMCTP4. To monitor recovery in cortical ER, and hence, ER-PM contacts, upon AtMCTP4 expression, we used Sec63-RFP [77] as an ER marker combined with confocal microscopy. In wild-type cells, the ER was organised into nuclear (nER) and cER. The cER was visible as a thread of fluorescence along the cell periphery, covering a large proportion of the cell circumference (Fig 8A white arrows). By contrast and as previously reported [76], we observed a substantial reduction of cER in the Δtether mutant, with large areas of the cell periphery showing virtually no associated Sec63-RFP (Fig 8A). When AtMCTP4 was expressed into the Δtether mutant line, we observed partial recovery of cER, visible as small regions of Sec63-RFP closely apposed to the cell cortex. We further quantified the extent of cER in the different lines by measuring the ratio of the length of cER (Sec63-RFP) against the cell perimeter (through

**Figure 7.  Anionic lipid-dependent membrane docking of AtMCTP4 C2 domains.**

A    Top: 3D-atomistic model of the individual AtMCTP4 C2 domains. Beta strands are shown in pink, loops in green and alpha helices in orange. Bottom: molecular dynamics of individual AtMCTP4 C2 domains with different biomimetic lipid bilayer compositions: phosphatidylcholine (PC) alone, with phosphatidylserine (PS) (PC/PS 3:1) and with PS, sitosterol (Sito) and phosphoinositol-4-phosphate (PI4P)(PC/PS/Sito/PI4P 57:19:20:4). The plots show the distance between the protein's closest residue to the membrane and the membrane centre, over time. The membrane's phosphate plane is represented by a $PO_4$ grey line on the graphs and a dark green meshwork on the simulation image captures (above graphs). For each individual C2 domain and a given lipid composition, the simulations were repeated four to five times (runs 1–5). C2 membrane docking was only considered as positive when a minimum of four independent repetitions showed similarly stable interaction with the membrane. All C2 domains of AtMCTP4 show membrane interaction when anionic lipid, in particular PI4P, is present. The amino acid colour code is as follows: red, negatively charged (acidic) residues; blue, positively charged (basic) residues; green, polar uncharged residues; and white, hydrophobic residues.

B, C   Confocal microscopy of *Arabidopsis* root epidermal cells of UBQ10:YFP-AtMCTP4 transgenic lines after 40-min treatment with DMSO (mock) and PAO (60 μM), an inhibitor of PI4 kinase. To confirm PI4P depletion upon PAO treatment, we used the PI4P *Arabidopsis* sensor line 1xPH(FAPP1) [74]. (B) PAO treatment leads to a loss of plasmodesmal punctate signal at the cell periphery (apical–basal boundary is highlighted by white arrowheads in B) for YFP-AtMCTP4, and redistribution of PM-localised 1xPH(FAPP1) to the cytoplasm. (C) Magnified boxed regions from (B) and profile plot along the cell wall after DMSO (1) or PAO (2) treatment, respectively (arrows: plasmodesmal punctae). Scale bars, 5 μm in (B) and 2.5 μm in (C).

▶

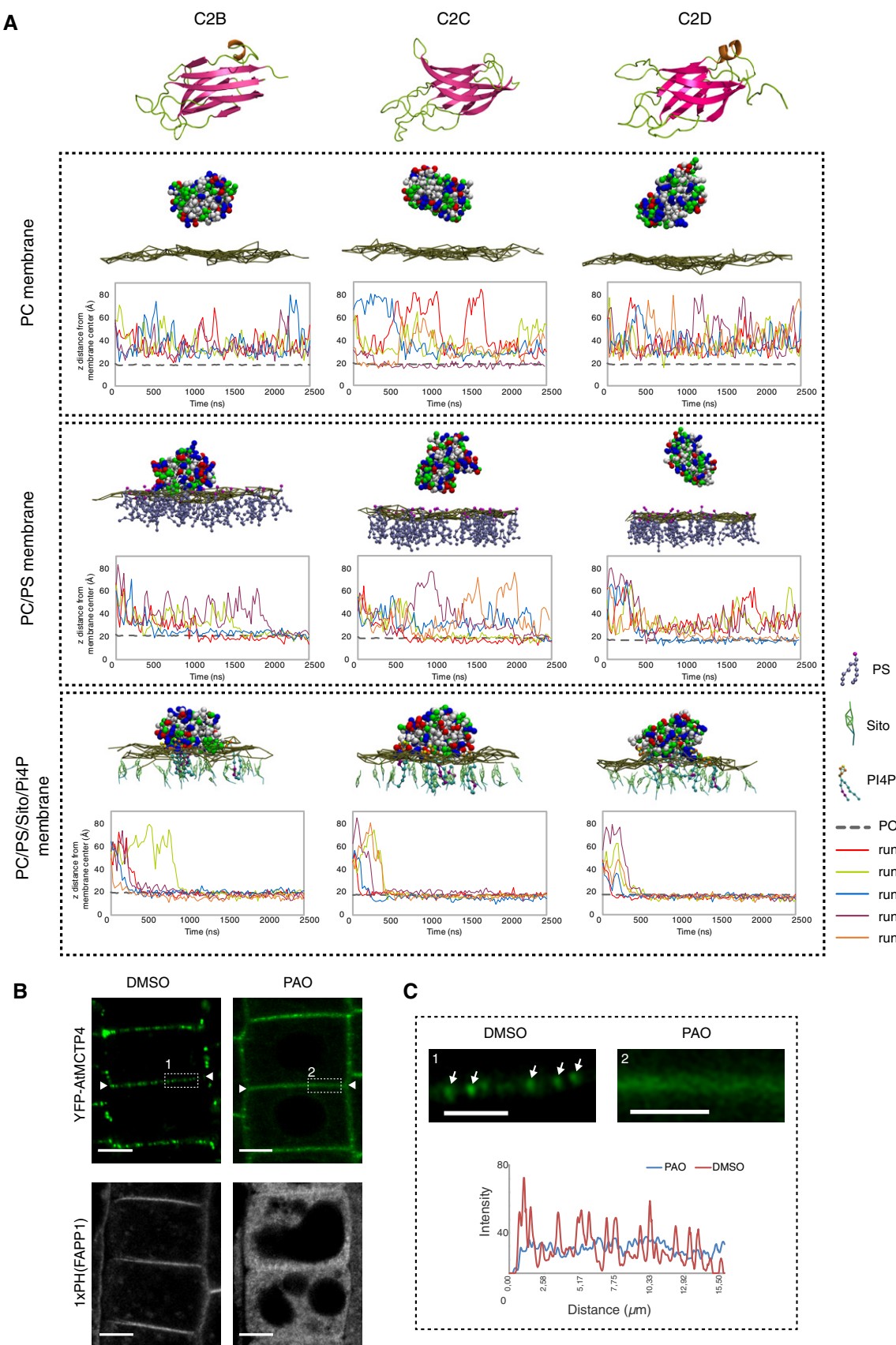

**Figure 7.**

calcofluor wall staining) and confirmed that AtMCTP4 expression induced an increase in cER from 7.3 to 23.1% when compared to the Δtether mutant (Fig 8B). This partial complementation is similar to results obtained with yeast deletion mutants containing only a single endogenous ER-PM tether, IST2, or all three isoforms of the tricalbin (yeast homologs of HsE-Syts) [76], supporting a role of AtMCTP4 as ER-PM tether.

## Discussion

In plants, communication between cells is facilitated and regulated by plasmodesmata, ~50-nm-diameter pores that span the cell wall and provide cell-to-cell continuity of three different organelles: the PM, cytoplasm and ER. The intercellular continuity of the ER and the resulting architecture of the pores make them unique amongst eukaryotic cellular junctions and qualify plasmodesmata as a

specialised type of ER-PM MCS [1,33]. Like other types of MCS, the membranes within plasmodesmata are physically connected, but so far, the molecular components and function of the ER-PM tethering machinery remain an enigma.

Here, we provide evidence that members of the MCTP family, some of which have been described as key regulators of intercellular trafficking and cell-to-cell signalling [20,51,52], also act as ER-PM tethers inside the plasmodesmata pores.

### MCTPs are functionally important plasmodesmal components

To identify ER-PM tether candidates acting specifically at plasmodesmata, we produced a refined plasmodesmata proteome. We combined label-free proteomic analysis and subcellular fractionation, and reduce the complexity of the plasmodesmata proteome from about 1,400 proteins [49] to 115 proteins. For that, we purified plasmodesmata from *Arabidopsis* liquid cell culture,

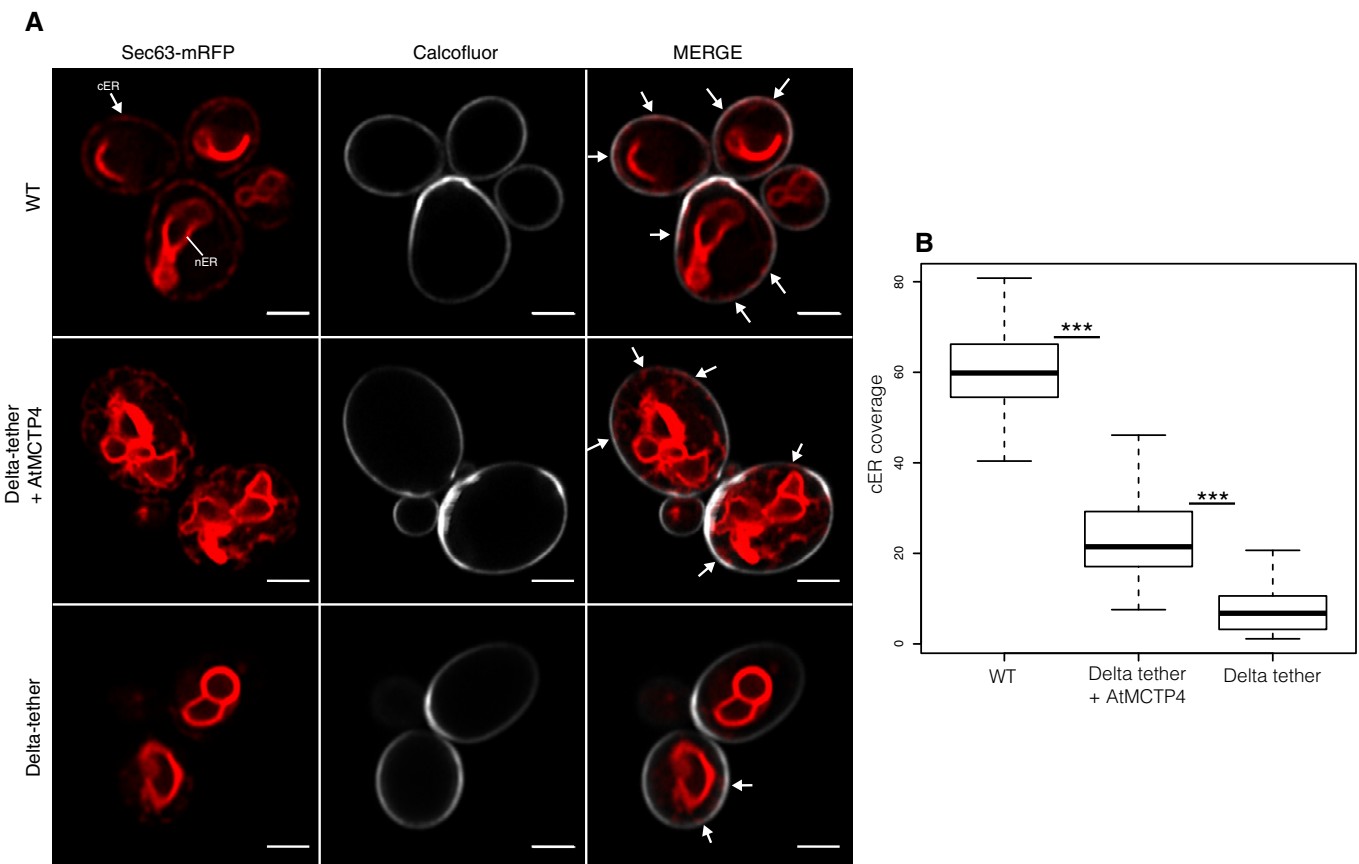

**Figure 8. AtMCTP4 expression in yeast partially restores ER-PM membrane contact sites.**

Expression of AtMCTP4 in yeast Δtether cells (ist2Δ, scs2/22Δ and tcb1/2/3Δ) [76] followed by confocal microscopic analysis of cortical ER.

A   Top to bottom: Wild-type (WT) cell, Δtether expressing untagged AtMCTP4 and Δtether cells, respectively. The cortical ER (cER) and nuclear ER (nER) are labelled by the ER marker Sec63-RFP (red), while the cell periphery is stained by calcofluor (white). In WT cells, both nER and cER are visible, whereas in Δtether cell only remains of the cER are visible (arrows), due to the loss of ER-PM tethering factors. When AtMCTP4 is expressed in the yeast Δtether, partial recovery of cER is observable (arrows). Scale bars, 2 μm.

B   Quantification of cER expressed as a ratio of the length of cER to length of the PM in WT, Δtether+AtMCTP4 and Δtether cells. Numbers of cells used for quantifying the cER: *n* = 39 for WT, *n* = 49 for Δtether+AtMCTP4 and *n* = 61 for Δtether strains. Wilcoxon test was used to compare the extent of cER between the different strains, i.e. WT versus Δtether+AtMCTP4 and Δtether+AtMCTP4 versus Δtether (***P < 0.001). In the box plot, median is represented by horizontal line, values between quartiles 1 and 3 are represented by box ranges, and minimum and maximum values are represented by error bars.

employing a protocol previously used to determine the lipid composition of these structures and which gives rise to pure plasmodesmata-derived membrane fractions [27,50,54]. We also took advantage of a label-free proteomic approach to simultaneously analyse plasmodesmata, PM, microsomal, cell wall and total cell extracts with the aim to discriminate plasmodesmata constituents from potential contaminants. Compared to the previously published *Arabidopsis* plasmodesmata proteome [49], our refined proteomic analysis is more stringent yet includes most of the well-established plasmodesmata protein residents such as members of the PDLP, PDCB, β1-3 glucanase and callose synthase families [4,6,29,60,78,79] (Appendix Table S1).

Using this proteomic approach, we identified several members of the MCTP family as plasmodesmata-enriched constituents, which we confirmed by confocal analysis using fluorescently tagged protein fusion, immunogold labelling and CLEM.

So far, two members of the MCTP family, AtMCTP1/FTIP and AtMCTP15/QKY, have been conclusively identified as plasmodesmata-associated proteins in *Arabidopsis*, both proteins acting as regulators of cell-to-cell signalling [20,51]. A recent study in Maize [80] also reports the localisation of CPD33 (AtMCTP15/QKY homolog) in plasmodesmata and at the ER, with cdp33 loss of function mutant exhibiting defects in plasmodesmata-mediated carbohydrate distribution. Here, we localise several further family members to plasmodesmata, including some for which other subcellular localisations were previously reported [52,53]. In particular, a recent paper by Liu *et al* [52] identified AtMCTP3 and AtMCTP4 as endosomal but also golgi, plasma membrane and cytosolic proteins whereas our data indicate that both proteins are plasmodesmata and ER-located. We fused GFP or YFP to the N-terminus of AtMCTP3 and AtMCTP4. This is in contrast to Liu *et al* [52], who inserted either GFP or RFP internally within the coding sequence or a 4xHA tag at the N-terminus, and reported both fusions as complementing *Atmctp3/Atmctp4*. AtMCTP4 fused to GFP at the N-terminus localised to plasmodesmata in transient expression in *N. benthamiana* leaves [63]. It has previously been shown for AtMCTP15/QKY that a functional, N-terminal GFP fusion is located at plasmodesmata, a subcellular localisation supported by immunogold electron microscopy, whereas a non-functional, C-terminal fusion shows a PM localisation [20,81]. On the other hand, a C-terminal GFP fusion and a N-terminal 4xHA tag fusion of AtMCTP1/FTIP were both located at plasmodesmata and the ER and functional [51]. Internally [52] or N-terminally (this study) fused AtMCTP constructs both complemented *Atmctp3/Atmctp4* double mutant. Thus, fusions at different positions may affect localisation of various MCTPs differently, and the proteins may also function at more than one subcellular localisation. Similarly to Liu *et al* [52], we found that a *Atmctp3/Atmctp4* loss-of-function *Arabidopsis* mutant displays severe developmental defects, which include stem cell specification defects in the shoot, but also in the root which had not been investigated by Liu *et al* [52]. We further show that the double At*mctp3/Atmctp4* mutant is impaired in plasmodesmata trafficking, with reduced size exclusion limit, and has an altered plasmodesmal proteome.

## MCTPs as plasmodesmata-specific ER-PM tethers

While ER-PM contacts within plasmodesmata have been observed for decades [22–24,82], the molecular identity of the tethers has remained elusive. Here, we propose that MCTPs are prime plasmodesmal membrane tethering candidates as they possess all required features: (i) strong association with plasmodesmata; (ii) structural similarity to known ER-PM tethers such as HsE-Syts and AtSYTs [57,58,66] with an ER-inserted TMR at one end and multiple lipid-binding C2 domains at the other for PM docking; and (iii) the ability to partially restore ER-PM tethering in a yeast Δtether mutant.

Similarly to other ER-PM tethers [15,47,48,58], MCTP C2 domains dock to the PM through electrostatic interaction with anionic lipids, especially PI4P and to a lesser extent PS. In contrast with animal cells, PI4P is found predominantly at the PM in plant cells and defines its electrostatic signature [75]. Although PI4P depletion reduces AtMCTP4 association with the pores, it is unlikely that the lipid acts alone as a sole determinant of plasmodesmata targeting, as MCTP C2 domains without the TMR did not localise to the pores. Instead, a combination of protein/protein and protein/lipid interactions at both the ER and PM may collectively contribute to plasmodesmata targeting of the MCTP family. Although plasmodesmata are MCS, they are also structurally unique: both the ER and the PM display extreme, and opposing membrane curvature inside the pores; the ER tubule is linked to the PM on all sides; and the membrane apposition is unusually close (2–3 nm in type I post-cytokinetic pores [25]). Thus, while structurally related to known tethers, MCTPs are also expected to present singular properties. For instance, similar to the human MCTP2, plant MCTPs could favour ER membrane curvature through their TMR [83]. Plasmodesmata also constitute a very confined environment, which, together with the strong negative curvature of the PM, may require the properties of MCTP C2 domains to differ from that of HsE-Syts or AtSYTs. All of these aspects will need to be investigated in the future.

## Interorganellar signalling at the plasmodesmal MCS?

In yeast and animals, MCS have been shown to be privileged sites for interorganelle signalling by promoting fast, non-vesicular transfer of molecules such as lipids [15,40,68]. Unlike the structurally analogous tethering proteins AtSYTs and HsE-Syts, MCTPs do not harbour known lipid-binding domains that would suggest that they participate directly in lipid transfer between membranes. However, MCTPs are likely to act in complex with other proteins [81,84] which may include lipid-shuttling proteins. For instance, AtSYT1, which contains a lipid-shuttling SMP (synaptotagmin-like mitochondrial-lipid-binding protein) domain [85], is recruited to plasmodesmata during virus infection and promotes virus cell-to-cell movement [66]. MCS tethers typically interact with other MCS components and locally regulate their activity, act as Ca$^{2+}$ sensors or modulate membrane spacing to turn lipid shuttling on or off [37,38,41–46,58,68,86,87]. Similar activities could be performed by MCTPs at plasmodesmata and might contribute to the altered plasmodesmal proteome of the *Atmctp3.Atmctp4* mutant. To date however, ER-PM cross-talk at plasmodesmata remains hypothetical.

## Combining organelle tethering and cell-to-cell signalling functions

Several members of the MCTP family have previously been implicated in regulating either macromolecular trafficking or intercellular

signalling through plasmodesmata. AtMCTP1/FTIP interacts with and is required for phloem entry of the Flowering Locus T (FT) protein, triggering transition to flowering at the shoot apical meristem [51]. Similarly, AtMCTP3/AtMCTP4 regulate trafficking of SHOOTMERISTEMLESS in the shoot apical meristem; however, in this case they prevent cell-to-cell trafficking [52]. In this study, we have shown that an *Atmctp3.Atmctp4* mutant displays reduced macromolecular trafficking between leaf epidermal cells, though it remains to be investigated how this relates to trafficking of specific developmental signals. AtMCTP15/QKY promotes the transmission of an unidentified non-cell-autonomous signal through interaction with the plasmodesmata/PM-located receptor-like kinase STRUB-BELIG [20]. Thus, previously characterised MCTP proteins regulate intercellular trafficking/signalling either positively or negatively.

While the mechanisms by which these MCTP proteins regulate intercellular transport/signalling have not been elucidated, MCTP physical interaction with mobile factors or receptor is critical for proper function [20,51–53]. In AtMCTP1/FTIP, the interaction is mediated by the C2 domain closest to the TMR [53]. For the C2 domains of HsE-Syts, conditional membrane docking is critical for their function and depends on intramolecular interactions, cytosolic $Ca^{2+}$ and the presence of anionic lipids [58,68,87–89]. With three to four C2 domains, it is conceivable that MCTPs assume different conformations within the cytoplasmic sleeve in response to changes in the plasmodesmal PM composition, $Ca^{2+}$ and the presence of interacting mobile signals (Fig 9), which could link membrane tethering to cell-to-cell signalling. Understanding in detail how MCTPs function in the formation and regulation of the plasmodesmal MCS will be an area of intense research in the coming years.

## Materials and Methods

### Biological material and growth conditions

*Arabidopsis* (*Columbia*) and transgenic lines were grown vertically on solid medium composed of *Murashige and Skoog* (MS) medium including vitamins (2.15 g/l), MES (0.5 g/l) and plant agar (7 g/l), pH 5.7, and then transferred to soil under long-day conditions at 22°C and 70% humidity.

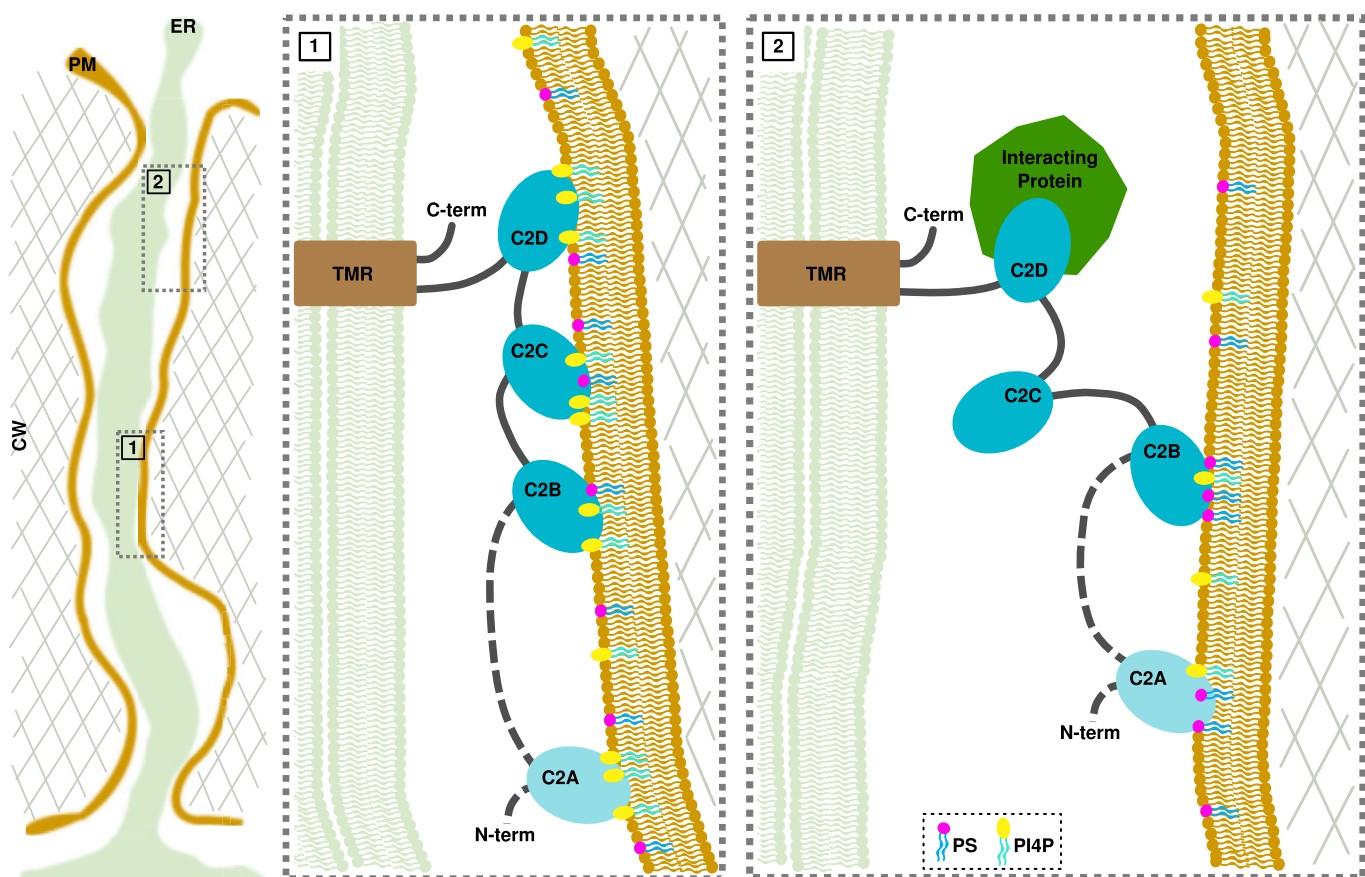

**Figure 9. Model of MCTP arrangement within plasmodesmata and hypothetical conditional docking events.**

Inside plasmodesmata, MCTPs insert into the ER via their transmembrane regions (TMR), while docking to the PM by interacting with the negatively charged phospholipids, PS and PI4P via their C2 domains. In condition of high PI4P/PS levels, all C2 domains interact with the PM, maintaining the ER close to the PM (panel 1). Decrease in the PI4P pool and/or protein interaction causes a detachment of some but not all C2 domains, which then modulate the space between the two membranes and the properties of the cytoplasmic sleeve. Please note that the exact topology of the TMR is not currently known.

*Arabidopsis* (*Landsberg erecta*) culture cells were cultivated as described in [Ref. 26] under constant light (20 µE/m/s) at 22°C. Cells were used for experimentation at various ages ranging from 4 to 7 days old (mentioned in individual experiment).

## MCTP sequence alignment and phylogenetic tree

The 16 members of *Arabidopsis thaliana* MCTP family, gathering a total of 59 C2 domains, were dissected using a combination of several bioinformatic tools. The alignment of *A. thaliana* MCTP members from [Ref. 53] combined with Pfam predictions was used as a first step to segregate the MCTP members into "subfamilies": the short MCTPs, which contain three C2 domains (C2B to C2D), and the long MCTPs, which contain four C2 domains (C2A to C2D). The short MCTPs lack the C2A domain, whereas the C2B, C2C and C2D are conserved in all members.

The prediction and delimitation of C2 domains in proteins, including MCTPs, from databases such as Pfam are rather imprecise. In order to provide stronger and more accurate predictions for the delimitation of each C2 domain, we used both the PSIPRED [90,91] protein sequence analysis (http://bioinf.cs.ucl.ac.uk/psipred/) and hydrophobic cluster analysis [92] (HCA; http://www-ext.impmc.upmc.fr/~callebau/HCA.html). Multiple sequence alignment was performed using Clustal Omega (http://www.ebi.ac.uk/Tools/msa/clustalo/).

## Cluster map of human and *A. thaliana* C2 domains

To generate a C2 cluster map, we first collected all *A. thaliana* and human C2 domains, using the HHpred web server [92,93]. The obtained set was filtered to a maximum of 100% pairwise sequence identity at a length coverage of 70% using MMseqs2 [95] to eliminate all redundant sequences. The sequences in the filtered set, comprising almost all human and *A. thaliana* C2 domains (~1,800 in total), were next clustered in CLANS [71] based on their all-against-all pairwise sequence similarities as evaluated by BLAST *P*-values.

## Cloning of MCTPs and transformation into *Arabidopsis*

The different constructs used in this study were either PCR amplified from cDNA or genomic DNA (Col-0) using gene-specific primers (Appendix Table S2), or were synthesised and cloned into donor vectors by GenScript® (Appendix Table S2). For N-terminal tag fusion, the PCR/DNA products were cloned into the MultiSite Gateway® donor vectors pDONR-P2RP3 (Invitrogen, Carlsbad, CA) and then subcloned into pB7m34GW or pK7m34GW using the multisite LR recombination system [96], the moderate promoter UBIQUITIN10 (UBQ10/pDONR-P4P1R previously described in [97]) and eYFP/pDONR221. For C-terminal tag fusion, the PCR/DNA products were first cloned into pDONR221, and then, was recombined using multisite recombinaison system using mVenus/pDONR-P2RP3 and UB10/pDONR-P4P1R.

For the expression of GFP-AtMCTP4 driven by its native promotor, we used the binary vector pRBbar-OCS harbouring a BASTA resistance, a multiple cloning side (MCS) and an octopine synthase (OCS) terminator within the left and right borders. The vector is derived from the pB2GW7 [96] by cutting out the expression cassette with the restriction enzymes SacI and HindIII and replacing it with a synthesised MCS and an OCS terminator fragment. To combine promoter region and GFP-AtMCTP4 coding sequence, we used In-Fusion Cloning (Takara Bio Europe). To PCR amplify the coding sequence for GFP-AtMCTP4 with its respective primers (Appendix Table S2), we used the plasmid coding for GFP-AtMCTP4 as template (previously described as GFP-C2-89 by [63]). The resulting pRBbar-pAtMCTP4: plasmid was linearised with BamH1/Pst1, and the amplified GFP-MCTP4 was fused in to generate the MCTP4 promoter-driven GFP-AtMCTP4 construct (pAtMCTP4:GFP-AtMCTP4).

Expression vectors were transformed in *Arabidopsis* Col-0 by floral dip [98], and transformed seeds were selected based on plasmid resistance.

*Nicotiana benthamiana* homologs of *Arabidopsis* MCTP isoforms were identified by protein BLAST searches against the SolGenomics *N. benthamiana* genome (https://solgenomics.net). An ortholog of AtMCTP7, NbMCTP7 (Niben101Scf03374g08020.1), was amplified from *N. benthamiana* leaf cDNA. The recovered cDNA of NbMCTP7 differed from the SolGenomics reference by the point mutation G287D and three additional silent nucleotide exchanges, as well as missing base pairs 1,678–1,716 which correspond to thirteen in-frame codons (encoding the amino acid sequence LKKEKFSSRLHLR). We note that this nucleotide and amino acid sequence is exactly repeated directly upstream (bp 1,639–1,677) in the SolGenomics reference and may thus represent an error in the *N. benthamiana* genome assembly. The recovered NbMCTP7 sequence has been submitted to database.

## Generation of *Atmctp3/Atmctp4* loss of function *Arabidopsis* mutant

*Atmctp3* (Sail_755_G08) and *Atmctp4* (Salk_089046) T-DNA insertional *Arabidopsis* mutants (background Col-0) were obtained from the *Arabidopsis* Biological Resource Center (http://www.arabidopsis.org/). Single T-DNA insertion lines were genotyped, and homozygous lines were crossed to obtain double homozygous *Atmctp3/Atmctp4*.

For genotyping, genomic DNA was extracted from Col-0, *Atmctp3* (GABI-285E05) and *Atmctp4* (SALK-089046) plants using chloroform:isoamyl alcohol (ratio 24:1), genomic DNA isolation buffer (200 mM Tris–HCL PH7.5, 250 mM NaCl, 25 mM EDTA and 0.5% SDS) and isopropanol. PCR was performed with primers indicated in Appendix Table S2. For transcript expression, total mRNA was extracted from Col-0 and *Atmctp3/Atmctp4* using RNeasy® Plant Mini Kit (Qiagen) and cDNA was produced using random and oligodT primers. The expression level of AtMCTP3, AtMCTP4 and ubiquitous Actin2 (ACT2) transcript was tested by PCR amplification using primers listed in Appendix Table S2.

## Macromolecular cell-to-cell trafficking assay

60 ng of a plasmid encoding GFP-sporamin under control of a 35S promoter [62] was mixed with 1-µm gold particles (Bio-Rad) suspended in ethanol and bombarded into fully expanded *Arabidopsis* leaf rosettes from approximately 1.5- to 2-cm distance using a home-built non-vacuum nitrogen pressure gun [99]. Leaves were detached and imaged at 72 h after bombardment. Clusters of fluorescent cells were counted manually on maximum projections.

## Label-free proteomic analysis of plasmodesmata

To establish the plasmodesmata core proteome, the identification and the relative amount of proteins in different cellular fractions, namely, the plasmodesmata, PM, total cell (TP) extract, microsomal and cell wall (CW) fractions, were determined with a label-free quantification method. Four to six biological replicates of each fraction were used for quantification. The plasmodesmata, PM and microsomal fractions were purified from liquid cell cultures of *Arabidopsis thaliana* (ecotype Landsberg *erecta*) as described in [Ref. 27,54] and the cell wall protein extract as in [Ref. 100].

Ten micrograms of each protein sample was solubilised in Laemmli buffer and deposited onto an SDS–PAGE gel for concentration and cleaning purposes. Separation was stopped after proteins entered the resolving gel, and following colloidal blue staining, the bands were excised and cut into $1\text{-mm}^3$ pieces. Gel pieces were destained in 25 mM ammonium bicarbonate and 50% acetonitrile (ACN), rinsed twice in ultrapure water and shrunk in ACN for 10 min. After ACN removal, gel pieces were dried at room temperature, covered with trypsin solution (10 ng/ml in 40 mM $NH_4HCO_3$ and 10% ACN), rehydrated at 4°C for 10 min and finally incubated overnight at 37°C. Gel pieces were then incubated for 15 min in 40 mM $NH_4HCO_3$ and 10% ACN at room temperature. The supernatant was collected, and a water:ACN:HCOOH (47.5:47.5:5) extraction solution was added to gel slices for 15 min. The extraction step was repeated twice. Supernatants were concentrated by vacuum centrifugation to a final volume of 100 μl and acidified. The peptide mixture was analysed with the UltiMate 3000 Nano LC System (Dionex, Amsterdam, The Netherlands) coupled to an Electrospray Q-Exactive quadrupole Orbitrap benchtop mass spectrometer (Thermo Fisher Scientific, San Jose, CA). Ten microlitres of peptide digests was loaded onto a 300-μm-inner diameter × 5-mm $C_{18}$ PepMap™ trap column (LC Packings) at a flow rate of 30 μl/min. The peptides were eluted from the trap column onto an analytical 75-mm id × 25-cm $C_{18}$ PepMap™ column (LC Packings) with a 4–40% linear gradient of solvent B in 48 min (solvent A was 0.1% formic acid in 5% ACN, and solvent B was 0.1% formic acid in 80% ACN). The separation flow rate was set at 300 nl/min. The mass spectrometer was operated in positive ion mode at a 1.8-kV needle voltage. Data were acquired using Xcalibur 2.2 software in a data-dependent mode. MS scans ($m/z$ 300–2,000) were recorded at a resolution of $R = 70,000$ (@ $m/z$ 200) and an AGC target of $10^6$ ions collected within 100 ms. Dynamic exclusion was set to 30 s, and top 15 ions were selected from fragmentation in HCD mode. MS/MS scans with a target value of $1 \times 10^5$ ions were collected with a maximum fill time of 120 ms and a resolution of $R = 35,000$. Additionally, only +2 and +3 charged ions were selected for fragmentation. Other settings were as follows: no sheath nor auxiliary gas flow, heated capillary temperature, 250°C; normalised HCD collision energy of 25%; and an isolation width of 3 $m/z$. Data were searched by SEQUEST through Proteome Discoverer 1.4 (Thermo Fisher Scientific) against a subset of the version 11 of the Araport (https://www.araport.org/) protein database (40,782 entries). Spectra from peptides higher than 5,000 Da or lower than 350 Da were rejected. The search parameters were as follows: the mass accuracy of the monoisotopic peptide precursor and peptide fragments was set to 10 ppm and 0.02 Da, respectively. Only b- and y ions were considered for mass calculation. Oxidation of methionines (+16 Da) was considered as variable modification and carbamidomethylation of cysteines (+57 Da) as fixed modification. Two missed trypsin cleavages were allowed. Peptide validation was performed using the Percolator algorithm [101], and only "high-confidence" peptides were retained, corresponding to a 1% false-positive rate at peptide level.

For label-free quantitative data analysis, raw LC-MS/MS data were imported in Progenesis QI for Proteomics 2.0 (Nonlinear Dynamics Ltd, Newcastle, U.K.). Data processing includes the following steps: (i) features detection; (ii) features alignment across the twenty-six samples; (iii) volume integration for two to six charge-state ions; (iv) normalisation on ratio median; (v) import of sequence information; and (vi) calculation of protein abundance (sum of the volume of corresponding peptides). Only non-conflicting features and unique peptides were considered for calculation at the protein level. Quantitative data were considered for proteins quantified by a minimum of two peptides. Protein enrichment ratios were calculated between each protein in the plasmodesmata fraction and the same protein in the four other cellular fractions. Before that, a relative normalised abundance was established for each protein to the most abundant protein in the fraction of interest. Protein enrichment was estimated by calculating the ratio between the relative normalised abundance of a given protein in the fraction of interest compared to other fractions.

Cut-offs for enrichment ratios were determined using a reference list of previously identified plasmodesmata proteins (see Appendix Table S1 with previously characterised plasmodesmal proteins outlined in orange). Cut-off scores of 8 for plasmodesmata/ PM, 40 for plasmodesmata/microsome, 30 for plasmodesmata/TP and 30 for plasmodesmata/CW were selected in order to filter out the false positives, as most well-established plasmodesmal proteins display similar or higher enrichment ratios. Please note that while LysM domain-containing GPI-anchored protein 2 (LYM2) has been characterised as a plasmodesmal protein [12], its enrichment ratio value was below our cut-off limit. We manually added this protein to the core plasmodesmata proteome.

ER proteomic dataset was extracted from [Ref. 55,56].

To estimate differential abundance of *Arabidopsis* plasmodesmal proteins type II (7-day-old cells) versus type I (4-day-old cultured cells), label-free proteomic analysis was carried out as described above. Four biological replicates for each condition were used. Enrichment ratios were calculated for individual protein in the seven (type II)- versus 4 (type I)-day-old plasmodesmata fraction. Cut-off absolute value of enrichment ratio was set at 1.3-fold. A *t*-test comparing type II protein-normalised abundance to type I protein-normalised abundance was established and the significant limit was fixed at 0.05 and below. Only proteins from the plasmodesmata core proteome, which responded to these two thresholds, were retained. Please note that differential accumulation of type I versus type II plasmodesmata in *Arabidopsis* cultured cells was established in previous work by Nicolas *et al* [25].

The mass spectrometry proteomics data have been deposited to the ProteomeXchange Consortium via the PRIDE [102] partner repository with the dataset identifier PXD006806.

## Comparative proteomic analysis

For comparative proteomic comparison, we used 5 biological replicates of *Atmctp3/Atmctp4* and Col-0 of fully developed leaves and

enriched plasmodesmata-containing cell wall as described by Kraner *et al* [63]. Following that protocol, the remaining proteins were solubilised and subsequently digested with trypsin. Desalted peptides were separated on a 160-min acetonitrile gradient by ultra-performance liquid chromatography (UPLC). After electron spray ionisation, samples were analysed by an Orbitrap Fusion Tribrid Mass Spectrometer in HCD fragmentation mode. Raw MS data files were analysed by using PEAKS Studio 8.5 (Bioinformatics Solutions, Waterloo, Ontario, Canada; [103]) against the *Arabidopsis* TAIR10 protein database (November 2010, 35 386 entries). For identification, we allowed parent mass tolerance of 10.0 ppm, fragment mass tolerance of 0.5 Da, and two missed cleavages. Carbamidomethylation of cysteines was set as static modification and oxidation of methionine as dynamic modification. For label-free quantification, the FDR threshold was set to 1% and retention time shift tolerance to 10 min.

## Confocal laser scanning microscopy

For transient expression in *N. benthamiana*, leaves of 3-week-old plants were pressure-infiltrated with GV3101 agrobacterium strains, previously electroporated with the relevant binary plasmids. Prior to infiltration, agrobacteria cultures were grown in Luria and Bertani medium with appropriate antibiotics at 28°C for 2 days, then diluted to 1/10 and grown until the culture reached an $OD_{600}$ of about 0.8. Bacteria were then pelleted and resuspended in water at a final $OD_{600}$ of 0.3 for individual constructs and 0.2 each for the combination of two. The ectopic silencing suppressor 19k was co-infiltrated at an $OD_{600}$ of 0.15. Agroinfiltrated *N. benthamiana* leaves were imaged 3–4 days post-infiltration at room temperature. ~ 2 by 2 cm leaf pieces were removed from plants and mounted with the lower epidermis facing up onto glass microscope slides.

Transgenic *Arabidopsis* plants were grown as described above. For primary roots, lateral roots and hypocotyl imaging, 6- to 7-day-old seedlings or leaves of 5- to 8-leaf stage rosette plants were mounted onto microscope slides. For shoot apical meristem imaging, the plants were first dissected under a binocular, then transferred to solid MS media and immediately observed using a long-distance working 40× water-immersion objective.

Confocal imaging was performed on a Zeiss LSM 880 confocal laser scanning microscope equipped with fast AiryScan using Zeiss C PL APO x63 oil-immersion objective (numerical aperture 1.4). For GFP, YFP and mVenus imaging, excitation was performed with 2–8% of 488 nm laser power and fluorescence emission collected at 505–550 nm and 520–580 nm, respectively. For RFP and mCherry imaging, excitation was achieved with 2–5% of 561 nm laser power and fluorescence emission collected at 580–630 nm. For aniline blue (infiltrated at the concentration of 25 μg/ml) and calcofluor white (1 μg/ml), excitation was achieved with 5% of 405 nm laser and fluorescence emission collected at 440–480 nm. For co-localisation, sequential scanning was systematically used.

For quantification of NbMCTP7 co-localisation with VAP27.1, SYT1 and PDCB1, co-expression of the different constructs was done in *N. benthamiana*. An object-based method was used for co-localisation quantification [104]. Images from different conditions are all acquired with same parameters (zoom, gain, laser intensity, etc.), and channels are acquired sequentially. These images are processed and filtered using ImageJ software (https://imagej.nih.gov/ij/)

in order to bring out the foci of the pictures. These foci were then automatically segmented by thresholding, and the segmented points from the two channels were assessed for co-localisation using the ImageJ plugin *Just Another Colocalization Plugin* (*JACoP*) [104]. This whole process was automatised using a macro (available upon demand).

Pseudo-Schiff propidium iodide-stained *Arabidopsis* root tips were performed according to [Ref. 104]. Aniline blue staining was performed according to [Ref. 27]. Brightness and contrast were adjusted on ImageJ software (https://imagej.nih.gov/ij/).

## Plasmodesmata (PD) index

Plasmodesmata depletion or enrichment was assessed by calculating the fluorescence intensity of GFP/YFP-tagged full-length MCTP, truncated MCTPs and the proton pump ATPase GFP-PMA2 [106], at (i) plasmodesmata (indicated by mCherry-PDCB1, PDLP1-mRFP or aniline blue) and (ii) the cell periphery (i.e. outside plasmodesmata pit fields). For that, confocal images of leaf epidermal cells (*N. benthamiana* or *Arabidopsis*) were acquired by sequential scanning of mCherry-PDCB, PDLP1-mRFP or aniline blue (plasmodesmata markers) in channel 1 and GFP/YFP-tagged MCTPs in channel 2 (for confocal setting, see above). About thirty images of leaf epidermis cells were acquired with a minimum of three biological replicates. Individual images were then processed using ImageJ by defining five regions of interest (ROI) at plasmodesmata (using plasmodesmata marker to define the ROI in channel 1) and five ROIs outside plasmodesmata. The ROI size and imaging condition were kept the same. The GFP/YFP-tagged MCTP mean intensity (channel 2) was measured for each ROI and then averaged for single image. The plasmodesmata index corresponds to intensity ratio between fluorescence intensity of MCTPs at plasmodesmata versus outside the pores. For the plasmodesmata index of RFP-HDEL, PDLP1-RFP and mCherry-PDCB1, we used aniline to indicate pit fields. R software was used for making the box plots and statistics.

## FRAP analysis

For FRAP analysis, GFP-NbMCTP7, RFP-HDEL and mCherry-PDCB1-expressing *N. benthamiana* leaves were used. The experiments were performed on a Zeiss LSM 880 confocal microscope equipped with a Zeiss C PL APO x63 oil-immersion objective (numerical aperture 1.4). GFP and mCherry were respectively excited at 488 nm and 561 nm with 2% of argon or DPSS 561-10 laser power, and fluorescence was collected with the GaAsp detector at 492–569 nm and 579–651 nm, respectively. To reduce as much as possible scanning time during FRAP monitoring, the acquisition window was cropped to a large rectangle of 350 by 50 pixels, with a zoom of 2.7 and pixel size of 0.14 μm. By this mean, pixel dwell time was of 0.99 μs and total frame scan time could be reduced down to 20 ms approximately. Photobleaching was performed on rectangle ROIs for the ER network and on circle ROIs for the pit fields with the exciting laser wavelengths set to 100%. The FRAP procedure was the following: 30 pre-bleach images, 10 iterations of bleaching with a pixel dwell time set at 1.51 μs and then 300 images post-bleach with the "safe bleach mode for GaAsp", bringing up the scan time up to approximately 200 ms. The recovery profiles were background substracted and then double normalised (according to the last

pre-bleach image and to the reference signal, in order to account for observational photobleaching) and set to full scale (last pre-bleach set to 1 and first post-bleach image set to 0), as described by Kote Miura in his online FRAP-teaching module (EAMNET-FRAP module, https://embl.de). Plotting and curve fitting were performed on GraphPad Prism (GraphPad Software, Inc.).

### 3D-SIM imaging

For 3D structured illumination microscopy (3D-SIM), an epidermal peal was removed from a GFP-NbMCTP7-expressing leaf and mounted in perfluorocarbon PP11 [106] under a high-precision (170 μm ± 5 μm) coverslip (Marie Enfield). The sample chamber was sealed with non-toxic Exaktosil N 21 (Bredent, Germany). 3D-SIM images were obtained using a GE Healthcare/Applied Precision OMX v4 BLAZE with a 1.42NA Olympus PlanApo N 60× oil-immersion objective. GFP was excited with a 488 nm laser and imaged with emission filter 504–552 nm (528/48 nm). SR images were captured using DeltaVision OMX software 3.70.9220.0. SR reconstruction, channel alignment and volume rendering were done using softWoRx V. 7.0.0.

### Immunogold labelling on high-pressure frozen *Arabidopsis* roots

pAtMCTP4:GFP-AtMCTP4 roots were high-pressure frozen, freeze-substituted and embedded into HM20 resin as described in [Ref. 27]. Immunogold labelling was performed on 90-nm sections with the following antibodies: polyclonal anti-GFP antibody (Invitrogen A-11122 and Torrey lines TP-401) diluted at 1:200. Antibody binding was detected with 5 or 10 nm gold-conjugated goat anti-rabbit antibodies diluted 1:40. Quantification of immunogold labelling was performed by counting gold particles in plasmodesmata, at the cell wall outside plasmodesmata and endomembrane/cytosolic compartments. The numbers of gold particles were then normalised to the relative area of each compartment (compartment area/total analysed area). Statistical analysis was performed with the software R using the non-parametrical Wilcoxon test ($n = 30$ images; a total of 298 gold particles were counted).

### Correlative light and electron microscopy (CLEM)

Cell wall purification from 10-day-old seedlings (pAtMCPT4:GFP-AtMCTP4) was performed according to [Ref. 108]. The purified walls were washed twice for 10 min with 0.2 μm filtered deionised water before observation by confocal and electron microscopy. Wall fragments were directly deposited onto electron microscopy grids (T-400 mesh Cu, EMS) filmed with 2% parladion and carbon-coated, incubated for 3 min before removing water excess. The grid was then mounted in deionised water between glass slide and coverslip for confocal microscopy observation. The acquisition was performed using a Zeiss LSM 880 confocal microscope. The cell wall fragments were detected at 20× magnification and imaged as *z*-stack at 63× magnification (1.4 N.A., C-Plan-Apochromat, oil-immersion objective) with a pinhole (airy) of 1, excitation and emission filters at 405 nm and 415–490 nm, and at 488 nm and 500–560 nm for calcofluor white and GFP, respectively. The imaged wall fragments were identified by their overall position and shape on the grid which was then

recovered, dried and negative stained with 2% (w/v) uranyl acetate for transmission electron microscopy observation. The data acquisition was done on a FEI Tecnai G2 Spirit TWIN TEM with axial Eagle 4K camera at different magnifications to identify the wall fragment previously observed by confocal before focusing onto the region of interest. Afterwards, the low magnification images of both confocal and electron microscopy were superimposed for correlation (Photoshop).

Subsequent immunogold labelling was combined with CLEM. This requires blocking cell walls with 5% Natural Donkey Serum (NDS) in Tris-buffered saline (TBS) 1× for 1 h before incubation overnight at 4°C with monoclonal mouse antibodies against callose (β-(1-3)-glucan antibody; Biosupplies, Parkville, Victoria, Australia) diluted at 1:20. Antibody excess was washed four times (10 min) with TBS 1×. For callose and PDCB1 co-labelling, the cell walls were then blocked a second time with 5% NDS in TBS 1× for 1 h before incubation overnight at 4°C with polyclonal rabbit anti-PDCB1 antibodies [27] diluted at 1:300. Antibody excess was washed four times (10 min) with TBS 1×. Antibody binding was detected by 5 nm diameter gold-conjugated goat anti-rabbit antibodies (PDCB1) diluted at 1:30 (BB international) incubated for 2 h or 10 nm diameter gold-conjugated goat anti-mouse antibodies for callose and 5 nm diameter gold-conjugated goat anti-rabbit antibodies for PDCB1, both diluted at 1:30 and incubated for 2 h at room temperature. The control conditions for each immunogold labelling were performed following the same protocol without primary antibody incubation. After immunogold labelling, the cell wall fragments were washed twice 10 min with 0.2 μm filtered deionised water before observation by confocal and electron microscopy.

### PAO treatment

Short-term phenylarsine oxide (PAO) treatment was performed on 7-day-old *Arabidopsis* seedlings expressing pAtMCTP4:GFP-AtMCTP4 grown on solid agar plates containing MS salt (2.2 g/l) supplemented with vitamins, 1% sucrose and MES (0.5 g/l) at pH 5.8. For PAO treatment, seedlings were transferred to liquid MS media containing 60 μM PAO for 30–40 min before imaging. Controls were performed by replacing PAO with DMSO.

Long-term PAO treatment was performed by growing *Arabidopsis* Col-0 onto solid agar plates containing MS salt (2.2 g/l) supplemented with vitamins, 1% sucrose, MES (0.5 g/l) and 1 or 10 μM PAO for 7 days before imaging.

### 3D structure modelling and molecular dynamic simulations

The delimitation of each individual C2 domain of the MCTP family members was done by combining PSIPRED [89,90] secondary structure prediction, hydrophobic cluster analysis (HCA) [91] and multiple sequence alignment tools (Clustal Omega) [108], allowing a better definition of structured domains [109]. C2 domains are 130- to 140-residue water-soluble domains characterised by two facing beta-sheets of each four beta strands, with a hydrophobic core and loops connecting the beta strands.

AtMCTP4 C2 domain models were obtained using the automated ROBETTA server [110]. The three domains were aligned and built by comparative modelling from parents extended-synaptotagmin 2 (PDB id: 4npj) for C2B and C2C and Munc13-1 (PDB id: 3kwu) for C2D.

AtMCTP15 C2A, C2C and C2D domain models were obtained using T-COFFEE multiple sequence alignment [111,112], which served as input for Modeller [113] for homology modelling, with human E3 ubiquitin-like ligase NEDD4-like protein (PDB id : 2nsq), Munc13-1 (PDB id : 3kwt) and E3 ubiquitin-protein ligase NEDD4 (PDB id: 3b7y) for C2A; human E3 ubiquitin-like ligase NEDD4-like protein (PDB id : 2nsq), human Intersectin 2 (PDB id: 3jzy) and extended-synaptotagmin 2 (PDB id: 4npj) for C2C; and human MCTP2 (PDB id : 2ep6), human Intersectin 2 (PDB id: 3jzy) and extended-synaptotagmin 2 (PDB id: 4npj) for C2D. AtMCTP15 C2B domain was modelled using ROBETTA server [110] with alignment and comparative building from parent *Arabidopsis thaliana* CAR4 (PDB id: 5a51).

NbMCTP7 C2A domain was modelled using T-COFFEE multiple sequence alignment [111,112], which served as input for Modeller [113] for homology modelling, with human MCTP2 (PDB id: 2ep6), human E3 ubiquitin-like ligase NEDD4-like protein (PDB id: 2nsq) and Munc13-1 (PDB id: 3kwt). NbMCTP7 C2B, C2C and C2D were aligned and built by comparative modelling using ROBETTA server [110] from parents, C2 domain-containing protein from putative elicitor-responsive gene (PDB id: 1wfj) for C2B, the first C2 domain of human myoferlin (PDB id: 2dmh) for C2C and extended-synaptotagmin 2 (PDB id: 4npj) for C2D. Either ROBETTA [110] server or Modeller [113] was used for C2 domains modelling, depending on the quality of the template alignment. All the obtained C2 models were quality-verified using ProSA-web server [114].

The structural models were then used for molecular dynamic simulations with GROMACS v5 software [115]. Atomistic simulations have been performed with the GROMOS96 54a7 force field [117–119]. The systems were first minimised by steepest descent for 5000 steps. Then, NVT and NPT equilibrations were carried on for 1 ns with the protein under position restraints. Production runs were performed for 50 ns. The systems were solvated with SPC water [120]. All simulations were performed with a 2-fs time step, a short-range electrostatic cut-off and a short-range van der Waals cut-off of 1.0 nm. Bond lengths were maintained with the LINCS algorithm [121]. A Verlet cut-off scheme was used. Particle mesh Ewald (PME) [121] was used for long-range electrostatics. Temperature coupling was set to 300 K using v-rescale algorithm [123] with $\tau T = 0.1$ ps. For the NPT equilibration and the production run, pressure coupling was set to 1 bar using isotropic Parrinello-Rahman [124] with $\tau P = 2$ ps and compressibility at $4.5 \times 10^{-5}$ (bar$^{-1}$). Atomistic simulations showed that the characteristic beta sheet structure of C2 domains was stable along the 50-ns trajectories (RMSD < 0.15 nm) while the loops presented a greater mobility.

For the simulation of protein–membrane interactions, the models were converted to a coarse-grained (CG) representation suitable for the MARTINI 2.1 force field [125] with ELNEDIN [126] elastic network. To render the protein behaviour in CG, the ELNEDIN network was trimmed off at the high-mobility loop regions using the dom_ELNEDIN.tcl script [127] in VMD software [128]. Behaviour validation was performed by comparing the RMSF from the 50-ns atomistic simulation to a mean RMSF over three 50-ns CG simulations.

The CG structures were placed above PLPC (1-palmitoyl,2-linoleoyl-sn-glycero-3-phosphocholine), PLPC:PLPS (1-palmitoyl,2-linoleoyl-sn-glycero-3-phosphoserine) (3:1) or PLPC:PLPS:Sitosterol:PI4P (1-palmitoyl,2-linoleyl-sn-glycero-3-phosphoinositol-4-phosphate)

(57:19:20:4) bilayers built with the insane tool [129]. A 2000-step steepest descent energy minimisation and an equilibration of 1 ns with Berendsen [130] pressure coupling were carried out, followed by five production repetitions of 2.5 μs with a 20-fs time step. Temperature and pressure were coupled at 300 K and 1 bars using the v-rescale [123] and Parrinello-Rahman [124] algorithm, respectively, with $\tau T = 1$ ps and $\tau P = 12$ ps. Pressure was coupled semi-isotropically in XY and Z. A Verlet cut-off scheme was used, with a buffer tolerance of 0.005. Electrostatic interactions were treated with a reaction field, a Coulomb cut-off of 1.1 nm and dielectric constant of 15. van der Waals interactions; had a cut-off of 1.1 nm; and used a potential shift Verlet modifier [131]. Bond lengths were maintained with the LINCS algorithm [121].

The trajectories were analysed with the GROMACS v5 tools as well as with homemade scripts and MDAnalysis software [132,133]. 3D structures were analysed with both PyMOL (DeLano Scientific, http://www.PyMOL.org) and VMD softwares.

Once the proteins were in interaction with the lipid bilayer, the systems were transformed to an atomistic resolution with backwards [134,135]. Atomistic simulations have been performed with the GROMOS96 54a7 force field [117–119]. All the systems studied were first minimised by steepest descent for 5,000 steps. Then, NVT and NPT equilibrations were carried on for 0.1 and 1 ns, respectively. The protein was under position restraints, and periodic boundary conditions (PBC) were used with a 2-fs time step. Production runs were performed for 100 ns. Temperature was maintained by using the Nose–Hoover method [135] with $\tau T = 0.2$ ps during equilibration processes and v-rescale [123] with $\tau T = 2.0$ ps for production run. All the systems were solvated with SPC water [120], and the dynamics were carried out in the NPT conditions (300 K and 1 bar). A semi-isotropic pressure was maintained by using the Parrinello-Rahman barostat [124] with a compressibility of $4.5 \times 10^{-5}$ (1/bar) and $\tau P = 1$ ps. Verlet cut-off scheme was used for neighbour searching with fast smooth particle mesh Ewald (PME) [122] for electrostatics and twin range cut-offs for van der Waals interactions. Bond lengths were maintained with the LINCS algorithm [121].

**Yeast**

Wild-type (SEY6210) and delta-tether yeast strain [76] were transformed with Sec63.mRFP (pSM1959). Sec63.mRFP [77] was used as an ER marker and was a gift from Susan Mickaelis (Addgene plasmid #41837). Delta-tether/Sec63.mRFP strain was transformed with AtMCTP4 (pCU416 : pCU between SacI and SpeI sites, Cyc1 terminator between XhoI and KpnI sites and AtMCTP4 CDS between BamHI and SmaI sites, Appendix Table S2). Calcofluor white was used to stain the cell wall of yeast. All fluorescent microscopy was performed on midlog cells, grown on selective yeast media (-URA -LEU for AtMCTP4 and Sec63 expression, and -LEU for Sec63). Images were acquired with AiryScan module, using a 63× oil-immersion lens and sequential acquisition. Brightness and contrast were adjusted on ImageJ software (https://imagej.nih.gov/ij/).

Sequence data for genes in this article can be found in the GenBank/EMBL databases using the following accession numbers: AtMCTP1, At5g06850; AtMCTP2, At5g48060; AtMCTP3, At3g57880; AtMCTP4, At1g51570; AtMCTP5, At5g12970; AtMCTP6, At1g22610; AtMCTP7, At4g11610; AtMCTP8, At3g61300; AtMCTP9, At4g00700;

AtMCTP10, At1g04150; AtMCTP11, At4g20080; AtMCTP12, At3g61720; AtMCTP13, At5g03435; AtMCTP14, At3g03680; AtMCTP15, At1g74720; AtMCTP16, At5g17980 and NbMCTP7, Niben101Scf03374g08020.1.

## Data availability

Proteomic data: PRIDE PXD006800 (http://www.ebi.ac.uk/pride/archive/projects/PXD006800), PRIDE PXD006806 (http://www.ebi.ac.uk/pride/archive/projects/PXD006806) and PRIDE PXD013999 (http://www.ebi.ac.uk/pride/archive/projects/PXD013999).

Expanded View for this article is available online.

### Acknowledgements

This work was supported by the National Agency for Research (Grant ANR-14-CE19-0006-01 to E.M.B), "Osez l'interdisciplinarité" OSEZ-2017-BRIDGING CNRS programme to E.M.B., the European Research Council (ERC) under the European Union's Horizon 2020 research and innovation programme (grant agreement No 772103-BRIDGING) to E.M.B, the EMBO Young Investigator Program to E.M.B., and Fonds National de la Recherche Scientifique (NEAMEMB PDR T.1003.14, BRIDGING CDR J.0114.18 and RHAMEMB CDR J.0086.18) to L. L. and M.D. J.D.P. is funded by a PhD fellowship from the Belgian "Formation à la Recherche dans l'Industrie et l'Agriculture" (FRIA grant no. 1.E.096.18). Work in J.T. laboratory is supported by grant BB/M007200/1 from the U.K. Biotechnology and Biomedical Sciences Research Council (BBSRC) and by the Scottish Government's Rural and Environment Science and Analytical Services Division (RESAS). Fluorescence microscopy analyses were performed at the plant pole of the Bordeaux Imaging Centre (http://www.bic.u-bordeaux.fr). For electron microscopy, the Region Aquitaine also supported the acquisition of the electron microscope (grant no. 2011 13 04 007 PFM), and FranceBioImaging Infrastructure supported the acquisition of the AFS2 and ultramicrotome. The proteomic analyses were performed at the Functional Genomic Center of Bordeaux, (https://proteome.cgfb.u-bordeaux.fr). We thank Steffen Vanneste and Abel Rosado for providing the VAP27.1.RFP and SYT1.GFP binary vectors and Yvon Jaillais for providing the 1xPH(FAPP1) *Arabidopsis* transgenic lines. The plasmid pRBbar-OCS was kindly provided by Prof. Frederik Börnke (IGZ—Leibniz Institute of Vegetable and Ornamental Crops, Großbeeren, Germany). We thank Christophe Trehin and Patrice Morel for providing the AtMCT15_C2s construct and Alenka Copic for providing the yeast WT and Δtether strains. We thank Fabrice Cordelières for his help for the fluorescence image quantification and Paul Gouget, Yvon Jaillais, Andrea Paterlini and Yrjo Helariutta for critical review of the article prior to submission.

### Author contributions

FI, MSG, MF and SC carried out the label-free proteomic analysis of plasmodesmata fractions isolated from *Arabidopsis* cell cultures. MLB cloned the MCTPs, produced and phenotyped the *Arabidopsis* transgenic lines, with the exception of AtMCTP4:GFP-AtMCTP4 and 35S:GFP-AtMCTP6 which were generated by MK. MLB and JDP imaged the MCTP reporter lines. WJN carried out the FRAP analysis and image quantification for co-localisation with the help of LB. MG and JDP performed the CLEM and immunogold labelling approaches. JT, MR and VG performed the plasmodesmata cell-to-cell trafficking assay. MK carried out the proteomic comparison and AP the data integration. AG performed the phylogenic analysis. JDP and MLB carried out the PAO experiments. MLB performed the yeast experiments. TJH and JT performed the 3D-SIM. VA carried out the C2 cluster map analysis. JDP carried out the molecular dynamic analysis with the help of J-MC, LL and MD. EMB conceived the study and designed experiments with the help of JT and LL. EMB, JDP, JT, MLB and YH wrote the manuscript. All the authors discussed the results and commented on the manuscript.

### Conflict of interest

The authors declare that they have no conflict of interest.

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
