## [Review Process File · EMBO Reports]

Multiple C2 domains and Transmembrane region Proteins (MCTPs) tether membranes at plasmodesmata

Marie L. Brault, Jules D. Petit, Françoise Immel, William J. Nicolas, Marie Glavier, Lysiane Brocard, Amélia Gaston, Mathieu Fouché, Timothy J. Hawkins, Jean-Marc Crowet, Magali S. Grison, Véronique Germain, Marion Rocher, Max Kraner, Vikram Alva, Stéphane Claverol, Andrea Paterlini, Ykä Helariutta, Magali Deleu, Laurence Lins, Jens Tilsner, and Emmanuelle M. Bayer

Review timeline:

Submission date:	3 October 2018
Editorial Decision:	15 November 2018
Revision received:	15 March 2019
Editorial Decision:	8 May 2019
Revision received:	28 May 2019
Accepted:	6 June 2019

Transaction Report:

1st Editorial Decision

15 November 2018

Thank you for the submission of your research manuscript to our journal. We have now received the full set of referee reports that is copied below.

As you will see, the referee's opinion on the novelty and conclusiveness of the results are somewhat divergent, with referee 2 being most critical on both aspects. Yet, given the support from at least two referees, we would like to invite you to revise your manuscript for EMBO reports. I think that it will be critical to investigate the plasmodesmata phenotype of the *mctp3/4* double mutants in more detail and to test if the GFP-fusion constructs complement the mutant phenotype. It might also be advisable to complement the fluorescent imaging data with TEM and immunogold labeling techniques to some extent at least. The effect of PI4P on MCTP3/4 localization could also be investigated. Referee 2 indicated in his/her further feedback

" If the treatment induces MCTP3/4 dissipation from PD, the authors have a great opportunity to use that same treatment to examine PD phenotypes as well as plant phenotypes, linking plasmodesmal localization of MCTP3/4 with the double knock-out plant phenotypes. This would be a real excitement to the readers." These experiments would thus certainly strengthen the manuscript.

Thus, given these constructive comments, we would like to invite you to revise your manuscript with the understanding that the referee concerns (as detailed above and in their reports) must be fully addressed and their suggestions taken on board. Please address all referee concerns in a complete point-by-point response. Acceptance of the manuscript will depend on a positive outcome of a second round of review. It is EMBO reports policy to allow a single round of revision only and acceptance or rejection of the manuscript will therefore depend on the completeness of your responses included in the next, final version of the manuscript.

Revised manuscripts should be submitted within three months of a request for revision; they will otherwise be treated as new submissions. Please contact us if a 3-months time frame is not sufficient for the revisions so that we can discuss the revisions further.

Supplementary/additional data: The Expanded View format, which will be displayed in the main HTML of the paper in a collapsible format, has replaced the Supplementary information. You can submit up to 5 images as Expanded View. Please follow the nomenclature Figure EV1, Figure EV2 etc. The figure legend for these should be included in the main manuscript document file in a section called Expanded View Figure Legends after the main Figure Legends section. Additional Supplementary material should be supplied as a single pdf labeled Appendix. The Appendix includes a table of content on the first page with page numbers, all figures and their legends. Please follow the nomenclature Appendix Figure Sx throughout the text and also label the figures according to this nomenclature. For more details please refer to our guide to authors.

Regarding data quantification, please ensure to specify the name of the statistical test used to generate error bars and P values, the number (n) of independent experiments underlying each data point (not replicate measures of one sample), and the test used to calculate p-values in each figure legend. Discussion of statistical methodology can be reported in the materials and methods section, but figure legends should contain a basic description of n, P and the test applied. Please also include scale bars in all microscopy images.

We now strongly encourage the publication of original source data with the aim of making primary data more accessible and transparent to the reader. The source data will be published in a separate source data file online along with the accepted manuscript and will be linked to the relevant figure. If you would like to use this opportunity, please submit the source data (for example scans of entire gels or blots, data points of graphs in an excel sheet, additional images, etc.) of your key experiments together with the revised manuscript. Please include size markers for scans of entire gels, label the scans with figure and panel number, and send one PDF file per figure.

- a complete author checklist, which you can download from our author guidelines (<http://embor.embopress.org/authorguide#revision>). Please insert page numbers in the checklist to indicate where the requested information can be found.
 - a letter detailing your responses to the referee comments in Word format (.doc)
 - a Microsoft Word file (.doc) of the revised manuscript text
 - editable TIFF or EPS-formatted figure files in high resolution
- (In order to avoid delays later in the publication process please check our figure guidelines before preparing the figures for your manuscript:
http://www.embopress.org/sites/default/files/EMBOPress_Figure_Guidelines_061115.pdf)
- a separate PDF file of any Supplementary information (in its final format)
 - all corresponding authors are required to provide an ORCID ID for their name. Please find instructions on how to link your ORCID ID to your account in our manuscript tracking system in our Author guidelines (<http://embor.embopress.org/authorguide>).

As part of the EMBO publication's Transparent Editorial Process, EMBO reports publishes online a Review Process File to accompany accepted manuscripts. This File will be published in conjunction with your paper and will include the referee reports, your point-by-point response and all pertinent correspondence relating to the manuscript.

I look forward to seeing a revised version of your manuscript when it is ready. Please let me know if you have questions or comments regarding the revision.

REFeree REPORTS

Referee #1:

In the submitted manuscript titled "Multiple C2 domains and Transmembrane-region Proteins (MCTPs) tether membranes at plasmodesmata" by Brault et al. the authors present evidence that plant MCTPs are present at plasmodesmata (PD) and that they might have a function in connecting the plasmamembrane with the ER membrane in plasmodesmatal channels. Furthermore, they identified by proteomics and confirmed *in vivo* that some MCTPs are core plasmodesmal components - in particular *Nicotiana benthamiana* NbMCTP7, and *Arabidopsis* AtMCTP3, AtMCTP4, AtMCTP6, AtMCTP9, AtMCTP1/FTIP and AtMCTP15/QKY are found as PD associated. In addition, Brault et al. have very convincingly shown using fluorescent fusion proteins with predicted MCTP domains and C2 domains that these MCTPs associate to the plasma membrane and that the transmembrane region is associating to the ER. Finally they confirmed tether function of the most prominent PD localized AtMCTP4 in a yeast tether mutant by partial complementation. The summarized data provided by the authors is well beyond, more detailed and in-depth than that done previously for AtMCTP3 and AtMCTP4 recently reported by Liu et al, 2018. Notably, the new localization data in part disagree with the localization of these two proteins described by Liu et al. which is an important contribution to our understanding of their function in PD channels.

In general, the manuscript is well written and the presented data point to an important regulatory role of MCTPs in PD channels where the appressed ER and plasma membrane are in very close proximity and seem to be linked by MCTPs. Seemingly their function is in regulating transport of proteins between cells and tissues. These novel insights are an important contribution to our understanding of how transport via PD functions in plants. The experiments and approaches are well outlined and the presented data are of very high quality (especially the microscopic images showing the localization of the analyzed MCTPs). In my opinion the reported insights are an important step towards our understanding of how cell-to-cell signaling is established in plants.

Comment:

The authors should mention - or comment on - whether a complementation of *Atmctp3/Atmctp4* phenotype by the *pMCTP4:GFP-AtMCTP4*-fusion occurs. This would support their observation that the detected PD localization of the used artificial fusion protein is seen with a functional protein. This is especially important as N-terminal structural changes by the GFP fusion might significantly alter its subcellular presence in the ER and/or plasma membrane.

Minor suggestion:

- 1) Include the following sentence - or a similar statement - describing that FTIP and QUIRKY are reported MCTPs having a function in PD in the introduction text:
"AtMCTP1/Flower locus T Interacting Protein (FTIP) and AtMCTP15/QUIRKY (QKY) have previously been localised to plasmodesmata in *Arabidopsis* and implicated in cell-to-cell trafficking of developmental signals (Vaddepalli et al, 2014; Liu et al, 2012). However, two recent studies indicate that other MCTP members, including AtMCTP3, 4, 9, which show high plasmodesmata-enrichment in our proteome, do not associate with the pores *in vivo* (Liu et al, 2017, 2018)."
- 2) It would be worthwhile to include a short description in which tissues/cell types the *pMCTP4:GFP-AtMCTP4* fusion is present. E.g in cells having no function PD such as stomata, or cells having mature PD such a trichomes...etc.. and correlate it to the mutant phenotype.

Minor corrections:

"cell-cell signaling" should be changes to "cell-to-cell signaling" to distinguish apoplasmic from symplasmic signaling.

To define PD as singular cell junctions seems not to be appropriate as PD are true symplasmic

channels and not simple cell junctions as seen in animals.

Also the use of "flux" to describe macromolecular transport across PD seems to be odd as this is a biochemical term used for specific atoms/molecules passed through metabolic pathways.

Line 110 - correct "from cell-to-cell" to from cell to cell"

Line 864 and subsequent: correct missing spaces between text and citations such as "described in(Nicolas et al."

Inconsistent writing of mCHERRY vs. mCherry in text and figures.

Referee #2:

The manuscript submitted by Brault et al., reports characterization of a subset of Multiple C2 domains and Transmembrane region Proteins (MCTPs) family. Specifically, the authors describe subcellular localization and characterization of select MCTP members. This manuscript touches upon various aspects associated with MCTP, including subcellular localization of wild type and deletion mutant constructs as well as plant growth phenotyping of an Arabidopsis double knock-out mutant line and complementation of yeast mutant lacking a MCTP.

Overall, the results described are not coherent and the conclusions are insubstantial. Compared to previous papers describing specific MCTP members such as QKY and FTIP, nothing much new and exciting findings are described in this manuscript. Those published MCTP members have already been shown that they are associated with plasmodesmata, so it is not too surprising to learn that additional members also partially associate with plasmodesmata. Those members have also been shown to be involved in distinct developmental signaling pathways. In my opinion, the manuscript submitted by Brault et al., is missing key data that readers would expect to see: nailing down the localization of MCTP3/4 around and within plasmodesmata by showing immunogold labeling results and the functional meaning of their localization at plasmodesmata or giving mechanistic explanation to the plant phenotypes exhibited by *mctp3/4* mutants. The MCTP3/4 protein topology remains known but this information is critical for speculating how they may function as a membrane-tethered molecule. Hopefully, instead of touching on here and there without substantiating any of their findings, the authors refocus and address one or more key points when revising this manuscript. Below, I point out major issues I find in the manuscript.

Fig. 1: The authors claim that AtMCTP3-7, 9, 10, 14-16, are abundant and highly enriched at plasmodesmata and all MCTPs were more abundant (1.4 to 3.6 times) in type I (tight ER-PM contacts) compared with type II (open cytoplasmic sleeves) plasmodesmata, the classification that the group had used in their earlier publication. They introduce unconventional analysis such as "plasmodesmata index" to assert that MCTPs are ER proteins that are highly enriched at plasmodesmata-this is creative but misleading. This type of analysis-enrichment-requires a quantitative protein assay or immuno gold labeling. Moreover, the authors need to heed on the differences shown in subcellular distributions of their target proteins when they are transiently expressed in tobacco versus when expressed in stably transformed Arabidopsis using native promoters.

The authors show 3D-SIM analysis, again to assert the localization of MCTP within plasmodesmata, while noting the protein's association with the central cavity of plasmodesmal channel. To make such a claim, co-labeling of callose (or other known plasmodesmal marker) is required as reference to plasmodesmal orifices (see Fitzgibbon et al. 2019 for 3D-SIM).

Fig. 2: Here, authors use FRAP to assert that MCTP is stably associated or "clustered" at plasmodesmata. This is only a circumstantial data that does not tell anything about MCTP being associated with plasmodesmata. It only tells that MCTP "clusters" are less mobile than ER-partitioned from. The authors provide lengthy descriptions of data generated based on additional fluorescence imaging analysis and techniques, overselling the punctate fluorescence as critical evidence showing plasmodesmal localization. Yet, this question can be simply nailed down if authors present immunogold labeling data. What is the biological meaning of forming this clusters

or appearing as clusters at the cell periphery? Is this a natural phenomenon or an artifact due to a fluorescent tagging? In my opinion, all those fluorescent imaging analyses are mudding and distracting rather than reinforcing the point that authors aim to claim.

Fig. 3: The authors present data showing that growth and development of *Atmctp3/Atmctp4* double mutants are severely altered. However, this data does not help understanding the role of *AtMCTP3/AtMTCP4* nor connecting their localization information to the phenotype. Does GFP-tagged MCTP3 and MCTP4 complement the loss-of function *Atmctp3/Atmctp4* double mutants? Authors need to address this question because positive complementation result would support that the GFP-tagged form of MCTP3/4 is functional and that their subcellular localization and distribution is relevant to their function. Is plasmodesmal phenotype, either the channel morphology or cell-to-cell molecular movement, altered in this mutant? Is this mutant phenotype caused by alteration in plasmodesmal morphology induced by lack of MCTP3/4 function at plasmodesmata? Presenting the plant phenotype without addressing these questions is not useful nor helpful.

Fig. 4: To address indirectly the question of MCTP topology, the authors present ER-association of TMR truncated form of several MCTP members including previously reported FTIP and QKY. Vaddepalli et al (2014) have used a systematic mutagenesis approach creating a series of N- and C-terminal deletion mutants to be expressed under the control of the endogenous QKY promoter. To examine the role of each protein segment for subcellular targeting, they then introduced those constructs into *qky-8* background through stable transformation. They showed that QKY mutants are targeted to PM or ER depending on what parts are deleted. Their conclusion was that both specific C2 domains and PRT_C domain are required for plasmodesmal association. Here in this manuscript, the authors make a jumping conclusion that PRT_C domain is not sufficient to MCTPs' plasmodesmal association after seeing that PRT_C domain alone localizes to ER. Note that Vaddepalli et al. found other deletion constructs that retain PRT_C also localize to ER. Authors of this manuscript are recommended to carefully evaluate their conclusion (which is basically derived from one single deletion mutant) by producing comparably extensive mutagenesis data as exemplified by Vaddepalli et al.

I do not find clear significance of using yeast mutant complementation for a plasmodesmata-localized protein.

Referee #3:

Comments:

The authors have identified MCTP proteins that localize and function at the membrane contact sites between ER and PM of the plasmodesmata. The authors used cell fractionation, followed by proteomics and identified ~115 unique proteins, among them MCTPs, that are highly enriched in plasmodesmata. High resolution microscopy further confirmed plasmodesmata localization of the MCTPs. In addition, genetic analyses established functionality of MCTP3 and 4 in developmental processes, most notably abnormal phenotypes in root QC cells observed in double *mctp3/4* mutant lines. However, it is yet to be determined whether this apparent phenotype is caused by alteration of the structure of plasmodesmata or the size of the pores as the result of mutation in MCTP3/4 genes. Perhaps the authors could consider performing additional studies such as using 3D-SIM imaging or TEM to determine the size/structure of plasmodesmata in shoot or root apical meristem in the double mutant versus the respective wild type.

In addition, the authors claim that lipid composition, especially PI4P, guide the localization of MCTPs protein to the plasmodesmata. The even distribution of PI4P marker on plasma membrane, does not fully support the potential selective function of PI4P on targeting MCTPs specially/predominantly to plasmodesmata. As such, the central question remains to be addressed is how the MCTPs specific enrichment within plasmodesmata is achieved?

Overall this is an interesting paper that will benefit the field, provided the authors address the major aforementioned concerns.

Minor concerns:

1. In the abstract, the authors inaccurately noted that "plant have evolved unique MCS, the plasmodesmata intercellular pores, which combine.....". Perhaps the author could consider the following "the intercellular pores, plasmodesmata, evolved unique MCS, which combine.....".

2. Line 367, may be it would be more accurate to substitute cytosolic localization with endomembrane localization.

1st Revision - authors' response

15 March 2019

Referee #1:

In the submitted manuscript titled "Multiple C2 domains and Transmembrane-region Proteins (MCTPs) tether membranes at plasmodesmata" by Brault et al. the authors present evidence that plant MCTPs are present at plasmodesmata (PD) and that they might have a function in connecting the plasmamembrane with the ER membrane in plasmodesmatal channels. Furthermore, they identified by proteomics and confirmed *in vivo* that some MCTPs are core plasmodesmal components - in particular *Nicotiana benthamiana* NbMCTP7, and *Arabidopsis* AtMCTP3, AtMCTP4, AtMCTP6, AtMCTP9, AtMCTP1/FTIP and AtMCTP15/QKY are found as PD associated. In addition, Brault et al. have very convincingly shown using fluorescent fusion proteins with predicted MCTP domains and C2 domains that these MCTPs associate to the plasma membrane and that the transmembrane region is associating to the ER. Finally they confirmed tether function of the most prominent PD localized AtMCTP4 in a yeast tether mutant by partial complementation. The summarized data provided by the authors is well beyond, more detailed and in-depth than that done previously for AtMCTP3 and AtMCTP4 recently reported by Liu et al, 2018. Notably, the new localization data in part disagree with the localization of these two proteins described by Liu et al. which is an important contribution to our understanding of their function in PD channels.

In general, the manuscript is well written and the presented data point to an important regulatory role of MCTPs in PD channels where the appressed ER and plasma membrane are in very close proximity and seem to be linked by MCTPs. Seemingly their function is in regulating transport of proteins between cells and tissues. These novel insights are an important contribution to our understanding of how transport via PD functions in plants. The experiments and approaches are well outlined and the presented data are of very high quality (especially the microscopic images showing the localization of the analyzed MCTPs). In my opinion the reported insights are an important step towards our understanding of how cell-to-cell signaling is established in plants.

Comment:

The authors should mention - or comment on - whether a complementation of *Atmctp3/Atmctp4* phenotype by the pMCTP4:GFP-AtMCTP4-fusion occurs. This would support their observation that the detected PD localization of the used artificial fusion protein is seen with a functional protein. This is especially important as N-terminal structural changes by the GFP fusion might significantly alter its subcellular presence in the ER and/or plasma membrane.

Author's response: we thank the reviewer for positive feedback on our work. In the revised version of the manuscript we now provide data showing that AtMCTP3 (which shares 92.8% identity and 98.7% similarity in amino acids with AtMCTP4) N-terminal YFP fusion can indeed complement *Atmctp3/Atmctp4* phenotype (new Suppl. Figure 7).

Minor suggestion:

1) Include the following sentence - or a similar statement - describing that FTIP and QUIRKY are reported MCTPs having a function in PD in the introduction text:

"AtMCTP1/Flower locus T Interacting Protein (FTIP) and AtMCTP15/QUIRKY (QKY) have previously been localised to plasmodesmata in *Arabidopsis* and implicated in cell-to-cell trafficking of developmental signals (Vaddepalli et al, 2014; Liu et al, 2012). However, two recent studies indicate that other MCTP members, including AtMCTP3, 4, 9, which show high plasmodesmata-enrichment in our proteome, do not associate with the pores *in vivo* (Liu et al, 2017, 2018)."

Author's response: as suggested we have now included the above sentence in the introduction paragraph. Please note that we had already included similar statements at the relevant position in the results section.

2) It would be worthwhile to include a short description in which tissues/cell types the pMCTP4:GFP-AtMCTP4 fusion is present. E.g in cells having no function PD such as stomata, or cells having mature PD such a trichomes...etc.. and correlate it to the mutant phenotype.

Author's response: we have now included in the revised manuscript data on the localisation pattern of pAtMCTP4:GFP-AtMCTP4 at the organ/tissue level in *Arabidopsis*. Our data show strong expression of the protein fusion in the root tip including the QC but also in lateral root primordia, inflorescence meristem and in young leaves (Fig.5a; new Suppl. Figure 9a). We are now commenting about the significance of this with regards to the *Atmctp3/Atmctp4* developmental phenotypes.

We also provide localisation data of AtMCTP4 GFP fusion in leaves including stomata showing similar localization pattern as the PD-localised viral movement protein Cucumber mosaic virus (CMV) 3a (new Suppl. Figure 9b). Although guard cells have no plasmodesmata, there are still remnants i.e. half plasmodesmata of the epidermal side (see Wille, A.C., Lucas, W.J. (1984) Ultrastructural and histochemical studies on guard cells. *Planta* 160, 129-142), which are labelled by CMV 3a and AtMCTP4.

Minor corrections:

"cell-cell signaling" should be changes to "cell-to-cell signaling" to distinguish apoplasmic from symplasmic signaling.

Author's response: this point has been addressed

To define PD as singular cell junctions seems not to be appropriate as PD are true symplasmic channels and not simple cell junctions as seen in animals.

Author's response: we have replaced the term "cell junction" by "intercellular pore"

Also the use of "flux" to describe macromolecular transport across PD seems to be odd as this is a biochemical term used for specific atoms/molecules passed through metabolic pathways.

Author's response: we have replaced the term "flux" by "intercellular trafficking".

Line 110 - correct "from cell-to-cell" to from cell to cell"

Author's response: this point has been addressed

Line 864 and subsequent: correct missing spaces between text and citations such as "described in (Nicolas et al."

Author's response: this point has been addressed

Inconsistent writing of mCHERRY vs. mCherry in text and figures.

Author's response: we have corrected mCHERRY into mCherry throughout the manuscript

Referee #2:

The manuscript submitted by Brault et al., reports characterization of a subset of Multiple C2 domains and Transmembrane region Proteins (MCTPs) family. Specifically, the authors describe subcellular localization and characterization of select MCTP members. This manuscript touches upon various aspects associated with MCTP, including subcellular localization of wild type and deletion mutant constructs as well as plant growth phenotyping of an *Arabidopsis* double knock-out mutant line and complementation of yeast mutant lacking a MCTP.

Overall, the results described are not coherent and the conclusions are insubstantial. Compared to previous papers describing specific MCTP members such as QKY and FTIP, nothing much new and exciting findings are described in this manuscript. Those published MCTP members have already been shown that they are associated with plasmodesmata, so it is not too surprising to learn that additional members also partially associate with plasmodesmata. Those members have also been shown to be involved in distinct developmental signaling pathways. In my opinion, the manuscript submitted by Brault et al., is missing key data that readers would expect to see: nailing down the localization of MCTP3/4 around and within plasmodesmata by showing immunogold labeling results and the functional meaning of their localization at plasmodesmata or giving mechanistic explanation to the plant phenotypes exhibited by *mctp3/4* mutants. The MCTP3/4 protein topology remains known but this information is critical for speculating how they may function as a membrane-tethered molecule. Hopefully, instead of touching on here and there without substantiating any of their findings, the authors refocus and address one or more key points when revising this manuscript. Below, I point out major issues I find in the manuscript.

Author's response: we thank the reviewer for the critical analysis and constructive comments on the manuscript. Whilst FTIP and QKY have previously been shown to be plasmodesmata-localised and involved in regulating cell-to-cell signalling, we perceive the novelty and significance of our work to be that 1) we identify several MCTP family members as amongst the most abundant plasmodesmal proteins in a semi-quantitative proteomic analysis, implicating them as likely *structural* components; 2) we show that amongst these, several MCTPs previously described as located on endosomes are in fact located to plasmodesmata; 3) we link changes to plasmodesmata structure (Suppl Fig 2) and, in this revision, also function (see below) to abundance or presence of these proteins; and 4) we suggest for the first time that MCTPs may constitute, or be amongst the long-sought plasmodesmal membrane tethers ('spokes') and provide experimental and molecular modelling evidence to support this. Whilst components of signalling pathways are increasingly found at plasmodesmata (including the FTIP and QKY papers), to our knowledge structural components are still mostly unidentified, and we feel that this re-appreciation of the previously described MCTP proteins will constitute a significant advance for the field.

Fig. 1: The authors claim that AtMCTP3-7, 9, 10, 14-16, are abundant and highly enriched at plasmodesmata and all MCTPs were more abundant (1.4 to 3.6 times) in type I (tight ER-PM contacts) compared with type II (open cytoplasmic sleeves) plasmodesmata, the classification that the group had used in their earlier publication. They introduce unconventional analysis such as "plasmodesmata index" to assert that MCTPs are ER proteins that are highly enriched at plasmodesmata-this is creative but misleading. This type of analysis-enrichment-requires a quantitative protein assay or immuno gold labeling. Moreover, the authors need to heed on the differences shown in subcellular distributions of their target proteins when they are transiently expressed in tobacco versus when expressed in stably transformed Arabidopsis using native promoters.

Author's response: Given that plasmodesmata are extremely small sub-cellular structures, quantitative proteomics on highly purified plasmodesmal cell fractions is probably the most suitable quantitative protein assay available in this case. Quantitative analysis of intensities in fluorescence microscopy images is routinely used to gain information on relative protein abundances (e.g. Waters 2009 J Cell Biol doi:10.1083/jcb.200903097). We implemented the PD index as a practical way to showcase such difference in fluorescence intensity between MCTP localized to plasmodesmata and to non-plasmodesmata regions of the cell periphery. PD index has already been used in Perraki et al. PLoS Pathogen, 14(11): e1007378 (2018) (<https://doi.org/10.1371/journal.ppat.1007378>) to describe and quantify the extent of plasmodesmata association for a given protein. In the revised manuscript we performed immunogold labelling supporting MCTP enrichment at plasmodesmata. In addition we performed Correlative Light Electron Microscopy (CLEM) combined with immunogold labelling on purified walls to more clearly demonstrate plasmodesmata association at an ultrastructural level (new Figure 5b, d and Suppl Figure 11).

Concerning the transient vs stable expression of MCTP members, we see the same localization pattern, which is ER localization and an enrichment at plasmodesmata. We have however noticed that plasmodesmata enrichment is stronger in stable lines, which is something we commonly see for plasmodesmata markers, such as PDLP1 (see modified Suppl Figure 6).

The authors show 3D-SIM analysis, again to assert the localization of MCTP within plasmodesmata,

while noting the protein's association with the central cavity of plasmodesmal channel. To make such a claim, co-labeling of callose (or other known plasmodesmal marker) is required as reference to plasmodesmal orifices (see Fitzgibbon et al. 2019 for 3D-SIM).

Author's response: Our interpretation of the 3D-SIM images was based on the localisation of GFP signal within the dark cell wall area visible between adjacent cells. We have attempted dual-colour 3D-SIM imaging with an mCherry-plasmodesmal callose binding protein (PDCB) 1 fusion, an established plasmodesmata marker (Simpson et al 2009 doi:10.1105/tpc.108.060145), however this was unsuccessful due to the insufficient brightness of the RFP fusion.

In the revised manuscript, we now provide data showing AtMCTP4 localisation to plasmodesmata at the electron microscopic level using 1) CLEM on wall purified from pAtMCTP4:GFP-MCTP4 seedling and 2) Immunogold labelling against GFP on pAtMCTP4:GFP-MCTP4 root sections, showing anti-GFP gold-associated signal along the length of the plasmodesmata pores in support of the 3D-SIM data. (Figure 5).

Fig. 2: Here, authors use FRAP to assert that MCTP is stably associated or "clustered" at plasmodesmata. This is only a circumstantial data that does not tell anything about MCTP being associated with plasmodesmata. It only tells that MCTP "clusters" are less mobile than ER-partitioned from. The authors provide lengthy descriptions of data generated based on additional fluorescence imaging analysis and techniques, overselling the punctate fluorescence as critical evidence showing plasmodesmal localization. Yet, this question can be simply nailed down if authors present immunogold labeling data. What is the biological meaning of forming this clusters or appearing as clusters at the cell periphery? Is this a natural phenomenon or an artifact due to a fluorescent tagging? In my opinion, all those fluorescent imaging analyses are mudding and distracting rather than reinforcing the point that authors aim to claim.

Author's response: The motivation of our FRAP analysis was not to demonstrate plasmodesmata association, as this was already confirmed by co-localisation with known plasmodesmata markers (PDCB1, Figure 1; aniline blue, Figure 5 and suppl Figure 6). FRAP was used to compare the mobility of the protein at the plasmodesmata versus cortical ER. To answer the reviewer's concerns about the plasmodesmata localisation, we have performed both CLEM and immunogold labelling demonstrating localisation of AtMCTP4 GFP tag fusion with plasmodesmata pit fields (Figure 5c,d; Suppl Figure 11). However, we agree that the term "cluster" is misleading when referring to the FRAP. We have now removed the term "cluster" and replace "associate or localise"

Fig. 3: The authors present data showing that growth and development of *Atmctp3/Atmctp4* double mutants are severely altered. However, this data does not help understanding the role of AtMCTP3/AtMCTP4 nor connecting their localization information to the phenotype. Does GFP-tagged MCTP3 and MCTP4 complement the loss-of function *Atmctp3/Atmctp4* double mutants? Authors need to address this question because positive complementation result would support that the GFP-tagged form of MCTP3/4 is functional and that their subcellular localization and distribution is relevant to their function. Is plasmodesmal phenotype, either the channel morphology or cell-to-cell molecular movement, altered in this mutant? Is this mutant phenotype caused by alteration in plasmodesmal morphology induced by lack of MCTP3/4 function at plasmodesmata? Presenting the plant phenotype without addressing these questions is not useful nor helpful.

Author's response: We agree with the reviewer that the complementation of the double mutant by MCTP GFP tagged protein is important. We now provide data showing that the YFP-AtMCTP3 N-terminal fusion is able to complement the *Atmctp3/Atmctp4* mutant phenotype (suppl figure 7). We further show that the expression pattern of GFP-AtMCTP4 under native promoter at the tissue/organ level is consistent with the organs/tissues affected in the *Atmctp3/Atmctp4* phenotype, with strong expression is the root tip, young leaves and inflorescence shoot apical meristem (suppl figure 9). Lastly, in the revised manuscript, we provide evidence that both macromolecular cell-to-cell movement and plasmodesmata proteome are significantly altered in the *Atmctp3/Atmctp4* double mutant (Figure 4).

Fig. 4: To address indirectly the question of MCTP topology, the authors present ER-association of TMR truncated form of several MCTP members including previously reported FTIP and QKY.

Vaddepalli et al (2014) have used a systematic mutagenesis approach creating a series of N- and C-terminal deletion mutants to be expressed under the control of the endogenous QKY promoter. To examine the role of each protein segment for subcellular targeting, they then introduced those constructs into qky-8 background through stable transformation. They showed that QKY mutants are targeted to PM or ER depending on what parts are deleted. Their conclusion was that both specific C2 domains and PRT_C domain are required for plasmodesmal association. Here in this manuscript, the authors make a jumping conclusion that PRT_C domain is not sufficient to MCTPs' plasmodesmal association after seeing that PRT_C domain alone localizes to ER. Note that Vaddepalli et al. found other deletion constructs that retain PRT_C also localize to ER. Authors of this manuscript are recommended to carefully evaluate their conclusion (which is basically derived from one single deletion mutant) by producing comparably extensive mutagenesis data as exemplified by Vaddepalli et al.

Author's response: The aim of the deletion analysis carried out by Vaddepalli et al was to identify parts of QKY protein required for function (using endogenous promoter) and for plasmodesmata targeting (using ubiquitin promoter to obtain sufficient fluorescence signal). By contrast, we used N-terminally truncated MCTP proteins merely to show that their transmembrane regions are ER-associated, in full agreement with Vaddepalli et al's findings. Whilst these limited experiments are sufficient to conclude that the TMR (referred in the reviewer's comment as PRT_C domain) is not *sufficient* for plasmodesmal targeting, we of course do not conclude that it is not involved, as the cytoplasmic domain alone also does not localise to the channels. (We have modified the text in the result section to "*the TMR *alone* is not sufficient*" to make this less ambiguous).

The main purpose of our experiments was to evaluate the possibility for MCTPs to function as ER-PM tethering elements. We aimed at defining the role of two regions of the MCTP proteins, namely the TMR and the C2s block. Additional work will indeed be needed to further dissect the role of sub-segments of the protein, and the topology of the TMR, which we plan to address in future studies.

I do not find clear significance of using yeast mutant complementation for a plasmodesmata-localized protein.

Author's response: The aim of this assay was to experimentally test the ability of AtMCTP4 to physically tether ER and PM membranes *in vivo*. In our view, the fact that a plant MCTP protein can partially complement the yeast *Atether* mutant *despite* the absence of plasmodesmata in this heterologous organism actually emphasizes the membrane tethering ability of MCTPs.

Referee #3:

Comments:

The authors have identified MCTP proteins that localize and function at the membrane contact sites between ER and PM of the plasmodesmata. The authors used cell fractionation, followed by proteomics and identified ~115 unique proteins, among them MCTPs, that are highly enriched in plasmodesmata. High resolution microscopy further confirmed plasmodesmata localization of the MCTPs. In addition, genetic analyses established functionality of MCTP3 and 4 in developmental processes, most notably abnormal phenotypes in root QC cells observed in double *mctp3/4* mutant lines. However, it is yet to be determined whether this apparent phenotype is caused by alteration of the structure of plasmodesmata or the size of the pores as the result of mutation in MCTP3/4 genes. Perhaps the authors could consider performing additional studies such as using 3D-SIM imaging or TEM to determine the size/structure of plasmodesmata in shoot or root apical meristem in the double mutant versus the respective wild type.

Author's response: We thank the reviewer for her/his constructive comments. In the revised manuscript we now link the *Atmctp3/Atmctp4* phenotype with defects in plasmodesmata function. We provide data showing that the *Atmctp3/Atmctp4* mutant has reduced cell-to-cell movement compared to the wild type (Figure 4 a-b). We further show that the plasmodesmata proteome is significantly affected in the *Atmctp3/Atmctp4* double mutant (Figure 4 c-e). We are planning in further studies to determine whether plasmodesmata morphology and internal organisation is altered in the *Atmctp3/Atmctp4* mutant using electron tomography. Changes in plasmodesmal size exclusion limit caused by (non-tubule forming) viral movement

proteins do not correspond to significant morphological changes to the channels (e.g. Ding et al 1992 Plant Cell). Therefore, the effects of *Atmctp3/Atmctp4* knock out on plasmodesmata morphology may be quite subtle and we expect that it will only be meaningfully possible to investigate this with electron tomography. However, this work will require a significant investment in terms of time and will need to be performed at different cell type interfaces and developmental stages. We therefore think it is beyond the scope of this re-submission but this is definitely a point we want to address in future work.

In addition, the authors claim that lipid composition, especially PI4P, guide the localization of MCTPs protein to the plasmodesmata. The even distribution of PI4P marker on plasma membrane, does not fully support the potential selective function of PI4P on targeting MCTPs specially/predominantly to plasmodesmata. As such, the central question remains to be addressed is how the MCTPs specific enrichment within plasmodesmata is achieved?

Author's response: We do agree with the reviewer that PI4P alone cannot be a determinant for plasmodesmata localisation. We suspect that protein-protein interaction may also play an important role in addressing MCTP to plasmodesmata pores. We now discuss this point in the manuscript discussion.

Overall this is an interesting paper that will benefit the field, provided the authors address the major aforementioned concerns.

Minor concerns:

1. In the abstract, the authors inaccurately noted that "plant have evolved unique MCS, the plasmodesmata intercellular pores, which combine.....". Perhaps the author could consider the following "the intercellular pores, plasmodesmata, evolved unique MCS, which combine.....".

Author's response: We have replaced the sentence in the abstract by "*Plants have evolved a unique type of MCS, inside intercellular pores, the plasmodesmata, where endoplasmic reticulum (ER) - plasma membrane (PM) contacts coincide with regulation of cell-to-cell signalling*".

2. Line 367, may be it would be more accurate to substitute cytosolic localization with endomembrane localization.

Author's response: 1xPH(FAPP1) PI4P sensor has been reported before to indeed shift from PM to cytosolic localisation (Platre et al. 2018. Dev Cell. 45(4), P465-480).

2nd Editorial Decision

8 May 2019

Thank you for the submission of your revised manuscript to EMBO reports. I apologize again for the delay in handling your manuscript but we have only recently received the last referee report.

As you will see, referee 3 supports publication in EMBO reports without further revision, while referee 2 points out some remaining concerns. This referee is not fully convinced that MCTP4 is a core plasmodesmata constituent and points out that the localization contradicts an earlier report by Liu et al., 2018. I discussed this point further with referee 3, who considers the FP tagging and TEM data convincing. This reviewer noticed that you placed the GFP tag at the N-terminus in contrast to the C-terminal localization in Liu et al. These discrepancies and potential explanations for it should be further discussed in the final manuscript and clearly pointed out. The purity and specificity of plasmodesmata preps should be discussed as appropriate. Please also address the remaining referee concerns in a point-by-point response.

From the editorial side, there are also a few things that we need before we can proceed with the official acceptance of your study.

- Please provide a running title (max. 40 characters including spaces) and up to five keywords.

- Please specify the contributions of Yka Helariutta and Magali Deleu in the Author Contribution section
- You refer to "data not shown" on page 9 and 11. Please note that as per our editorial policies all data must be part of the manuscript. Therefore, please provide the relevant data.
- Please update the references to the numbered format of EMBO reports and rename the section "References". The abbreviation 'et al' should be used if more than 10 authors. You can download the respective EndNote file from our Guide to Authors
<https://drive.google.com/file/d/0BxFM9n2lEE5oOHM4d2xEbmpxN2c/view>
- Please add a callout to the figure panels of Figure 2 (Fig. 2a, b) on page 8.
- Movie 1 is never mentioned in the text.
- Movies:
 - (1) Please follow the nomenclature Movie EVx.
 - (2) Please provide their legend as separate plain README.txt file. Then zip movie and legend into one file and upload the zipped file.
- Supplement:
 - (1) Please note that all materials and methods must be part of the main manuscript file
 - (2) Please merge all other Supplemental figures and Supplemental tables into one .pdf file called Appendix and follow the nomenclature Appendix Figure Sx or Appendix Table Sx. The Appendix includes a table of content on the first page with page numbers, all figures and their legends.
- The accession number and database for the mass spec proteomics dataset should be listed in a formal "Data Availability" section (placed after Materials & Method) that follows the model below (see also <http://emboj.embopress.org/authorguide#dataavailability>).

Data availability

- [data type]: [name of the resource] [accession number/identifier/doi] ([URL or identifiers.org/DATABASE:ACCESSION])

- Figures:

- Fig 5A: the bottom right image is missing the magnification box.
- Fig 8A: please provide dividing lines between the boxes.
- Fig S5A - the top and middle merge boxes are incorrect. The same image is shown twice.
- Fig S6A, S7D: please indicate where the magnified images come from with a box.
- Fig S6B, legend: In the boxplots, please define the horizontal lines, box ranges and error bars. Please also indicate the number of replicates analysed.
- S7D, legend: please define the meaning of the arrows and the * in the legend.
- Fig S9B, legend: please define the meaning of the arrowheads and the size of the scale bar.
- Fig S12D, legend: In the boxplots, please define the horizontal lines, box ranges and error bars. Please also indicate the number of replicates analysed.
- Fig S12D, legend: please define the size of the scale bars.

- Our data editors from Wiley have already inspected the Figure legends for completeness and accuracy. Please see their suggested changes in the attached Word file.

- Finally, EMBO reports papers are accompanied online by A) a short (1-2 sentences) summary of the findings and their significance, B) 2-3 bullet points highlighting key results and C) a synopsis image that is 550x200-400 pixels large (width x height). You can either show a model or key data in

the synopsis image. Please note that the size is rather small and that text needs to be readable at the final size. Please send us this information along with the revised manuscript.

REFeree REPORTS

Referee #2:

I acknowledge that the authors have made huge extra efforts to address my comments.

Overall, the revised manuscript has many merits added to but in my opinion, the authors would want to tone down their claims and statements. For example, I do not agree that the authors can claim MCTP4 is a "core plasmodesmata-associated protein." I also do not find evidence that the authors could justify such mention in lines 494-6 that "...We therefore propose that MCTPs are core plasmodesmata constituents and that AtMCTP3 and AtMCTP4 may also regulate the transport of SHOOTMERISTEMLESS, at the pores." Their evidence presented is primarily based on PD proteomics and localization studies. However, it is not possible to prepare a PD-enriched fraction that is pure or near complete in composition using biochemical fractionation techniques currently available. Let's admit that. Using such a prep, abundance does not itself prove the protein being the most prominent PD component. Rather, it might represent most persistent contaminant or most well digested protein in the fraction, which produces abundant peptide peaks in MS analysis. If they are core structural components of PD, the consequence of their absence cannot be minor. Toning down the claim will not lose the significance of the function of the protein.

My real trouble comes from the discrepancy in results between the current manuscript and an earlier publication by Liu et al., which I was not aware of in terms of MCTP4 being a synonym for FTIP4 until I get to read that paper just recently. Liu et al. (Cell Rep. 2018) showed in their study that FTIP4, which corresponds to AtMCTP4, does not localize to PD but to intracellular vesicles. They demonstrated this using both FP tagging and immune-gold localization. This discrepancy is completely ignored in the current manuscript. Liu et al. showed that the fusion constructs they used for localization studies complemented *ftip3 ftip4* double mutants, which corresponds to *Atmctp3/4* double mutants. Thus, it is unlikely that the different conclusion on the FTIP4/MCTP4's subcellular localization patterns reported by Brault et al. from that by Liu et al. stems from different epitopes or FPs used in their studies. Liu et al. also showed that FTIP4 interacts with SHOOTMERISTEMLESS (STM) and STM trafficking is promoted in the absence of FTIP3/4 because STM partitioning is shifted more to the plasma membrane from nuclear localization. They explain in their paper that it is this change in STM activity, which brings about the stunted growth phenotypes of the *ftip3 ftip4* (*Atmctp3/4*) double mutants. Based on these data, if I understood correctly, it would make more sense if FTIP4/MCTP4 localizes to endosomes to perform its function regulating the partitioning of STM. If FTIP4/MCTP4 is a core PD component and most abundant at PD, how could STM normally move through PD at all?

Referee #3:

The additional data provided by the authors support their major conclusions, namely MCTPs located at plasmodesmata controlling ER-PM contacts and cell-to-cell signalling, and as such plant development. The authors have satisfactorily addressed my major concerns.

2nd Revision - authors' response

28 May 2019

Referee #2:

I acknowledge that the authors have made huge extra efforts to address my comments.

Overall, the revised manuscript has many merits added to but in my opinion, the authors would want to tone down their claims and statements. For example, I do not agree that the authors can claim MCTP4 is a "core plasmodesmata-associated protein." I also do not find evidence that the authors could justify such mention in lines 494-6 that "...We therefore propose that MCTPs are core plasmodesmata constituents and that AtMCTP3 and AtMCTP4 may also regulate the transport of SHOOTMERISTEMLESS, at the pores." Their evidence presented is primarily based on PD proteomics and localization studies. However, it is not possible to prepare a PD-enriched fraction that is pure or near complete in composition using biochemical fractionation techniques currently available. Let's admit that. Using such a prep, abundance does not itself prove the protein being the most prominent PD component. Rather, it might represent most persistent contaminant or most well digested protein in the fraction, which produces abundant peptide peaks in MS analysis. If they are core structural components of PD, the consequence of their absence cannot be minor. Toning down the claim will not lose the significance of the function of the protein.

We had referred to MCTP4 as a "core" plasmodesmata-associated protein, as it is amongst the most highly enriched proteins in the plasmodesmata fraction compared with other cellular compartments included in the semi-quantitative proteomic analysis, but have now removed the term "core" in most instances.

Obviously, cell fractionation and proteomic analyses are never entirely pure. But precisely for this reason we used a semi-quantitative proteomic approach that not only identifies the presence of candidate proteins, but provides a measure of their distribution among plasmodesmata and potentially contaminating fractions. A protein which is merely a prominent contaminant or digesting particularly well would be expected to show a high abundance also in at least some of the control fractions, and thus no significant enrichment in the plasmodesmata fraction. This approach is validated by the strong enrichment in the plasmodesmata fraction of previously identified, accepted plasmodesmal proteins. We have now added a paragraph in the discussion to discuss in more details the label-free proteomic facet of the paper.

With 16 members in the Arabidopsis MCTP family, we expect some redundancy, and different members may be important for plasmodesmata function in different tissues and at different developmental stages. However, in our view, the developmental phenotypes of the *Atmctp3/Atmctp4* mutant are consistent with a plasmodesmatal defect. We however agree that our study does not provide evidence for MCTP3/4 function in the transport of SHOOTMERISTEMLESS. We have therefore removed reference to SHOOTMERISTEMLESS in the revised manuscript.

My real trouble comes from the discrepancy in results between the current manuscript and an earlier publication by Liu et al., which I was not aware of in terms of MCTP4 being a synonym for FTIP4 until I get to read that paper just recently. Liu et al. (Cell Rep. 2018) showed in their study that FTIP4, which corresponds to AtMCTP4, does not localize to PD but to intracellular vesicles. They demonstrated this using both FP tagging and immune-gold localization. This discrepancy is completely ignored in the current manuscript. Liu et al. showed that the fusion constructs they used for localization studies complemented *ftip3 ftip4* double mutants, which corresponds to *Atmctp3/4* double mutants. Thus, it is unlikely that the different conclusion on the FTIP4/MCTP4's subcellular localization patterns reported by Brault et al. from that by Liu et al. stems from different epitopes or FPs used in their studies. Liu et al. also showed that FTIP4 interacts with SHOOTMERISTEMLESS (STM) and STM trafficking is promoted in the absence of FTIP3/4 because STM partitioning is shifted more to the plasma membrane from nuclear localization. They explain in their paper that it is this change in STM activity, which brings about the stunted growth phenotypes of the *ftip3 ftip4* (*Atmctp3/4*) double mutants. Based on these data, if I understood correctly, it would make more sense if FTIP4/MCTP4 localizes to endosomes to perform its function regulating the partitioning of STM. If FTIP4/MCTP4 is a core PD component and most abundant at PD, how could STM normally move through PD at all?

We have always referred to Liu et al. work and highlighted the discrepancy between their results and ours in terms of sub-cellular localisation (see lines 126-128; 198-199; 277; 314; and discussion). However to make this point even clearer, we have now expanded the comparison with Liu et al. 2018 in our discussion, and also included more detailed reference to fusion orientations for other previously characterised MCTP proteins.

Liu et al. report complementation of the *Atmctp3/Atmctp4* mutant by constructs with an internal G/RFP fusion, but also with a 4xHA tag fused to the N-terminus of MCTP4, which is the same orientation used in our G/YFP fusions. Thus, the complementation results are actually compatible. Our finding that loss of MCTP3/4, and thus, in our interpretation, less tight membrane tethering at plasmodesmata, leads to reduced macromolecular trafficking is counterintuitive, but is actually consistent with previous findings that immature plasmodesmata (with closer membrane contacts) are open for macromolecular trafficking (Nicolas *et al*, 2017) and in fact have larger size exclusion limits than 'open sleeve' plasmodesmata (Paper accepted for publication in Nature Plants). However, this does not necessarily imply that trafficking of STM should also be reduced, rather than increased, in *Atmctp3/Atmctp4*, as the presumably non-specific diffusion of 27 kDa GFP and the possibly regulated transport of 47 kDa STM may not be functionally equivalent.

Corresponding Author Name: Jens Tilsner and Emmanuelle Bayer

Journal Submitted to: EMBO report

Manuscript Number: EMBOR-2018-47182